# (De-)Randomized Smoothing for Decision Stump Ensembles

**Miklós Z. Horváth**,* **Mark Niklas Müller**,* **Marc Fischer, Martin Vechev**
Department of Computer Science
ETH Zurich
Switzerland
mihorvat@ethz.ch, {mark.mueller,marc.fischer,martin.vechev}@inf.ethz.ch

## Abstract

Tree-based models are used in many high-stakes application domains such as finance and medicine, where robustness and interpretability are of utmost importance. Yet, methods for improving and certifying their robustness are severely under-explored, in contrast to those focusing on neural networks. Targeting this important challenge, we propose deterministic smoothing for decision stump ensembles. Whereas most prior work on randomized smoothing focuses on evaluating arbitrary base models approximately under input randomization, the key insight of our work is that decision stump ensembles enable exact yet efficient evaluation via dynamic programming. Importantly, we obtain deterministic robustness certificates, even jointly over numerical and categorical features, a setting ubiquitous in the real world. Further, we derive an MLE-optimal training method for smoothed decision stumps under randomization and propose two boosting approaches to improve their provable robustness. An extensive experimental evaluation on computer vision and tabular data tasks shows that our approach yields significantly higher certified accuracies than the state-of-the-art for tree-based models. We release all code and trained models at `https://github.com/eth-sri/drs`.

## 1 Introduction

Tree-based models have long been a favourite for making decisions in high-stakes domains such as medicine and finance, due to their interpretability and exceptional performance on tabular data [1]. However, recent results have highlighted that tree-based models are, similarly to other machine learning models [2, 3], also highly susceptible to adversarial examples [4–6], raising concerns about their use in high-stakes domains where errors can have dire consequences.

While the robustness of neural models has received considerable attention [7–21], the challenge of obtaining robustness guarantees for ensembles of tree-based models has only been investigated recently [4, 22, 23]. However, these initial works only consider numerical features and are based on worst-case approximations, which do not scale well to the difficult $\ell_p$-norm setting.

**This Work** In this work, we address this challenge and present DRS, a novel (**De-**)**R**andomized **S**moothing approach, for constructing robust tree-based models with deterministic $\ell_p$-norm guarantees while supporting both categorical *and* numerical variables. Unlike prior work, our method is based on Randomized Smoothing (RS) [24], an approach that obtains robustness guarantees by evaluating a general base model under an input randomization $\phi(\boldsymbol{x})$. However, in contrast to standard applications of RS, which use costly and imprecise approximations via sampling and only obtain probabilistic certificates, we leverage the structure of decision stump ensembles to compute their exact output

---

*Equal contribution

36th Conference on Neural Information Processing Systems (NeurIPS 2022).

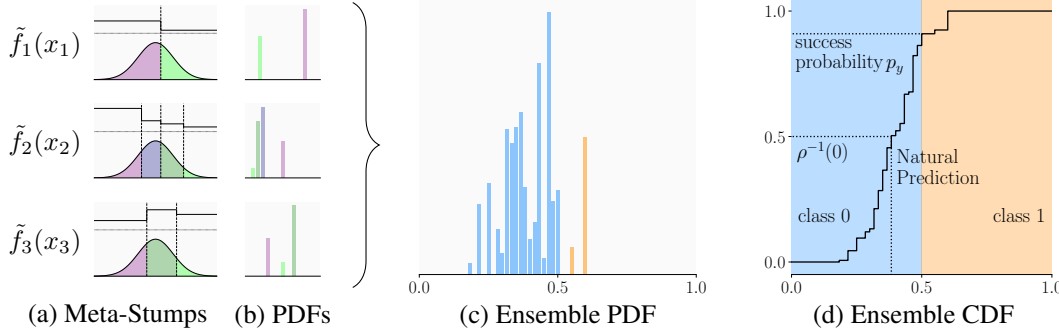

| (a) Meta-Stumps | (b) PDFs | (c) Ensemble PDF | (d) Ensemble CDF |

Figure 1: Given an ensemble of 3 meta-stumps $\tilde{f}_i$ (piecewise constant univariate functions), each operating on a different feature $x_i$ of an input $\boldsymbol{x}$, we calculate the probability of every output under input randomization (a) to obtain a distribution over their outputs (b). We aggregate these individual PDFs via dynamic programming to obtain the probability distribution over the ensemble's outputs (c). We can then compute the corresponding CDF (d) to evaluate the smoothed stump ensemble exactly.

distributions for a given input randomization scheme and thus obtain deterministic certificates. Our key insight is that this distribution can be efficiently computed by aggregating independent distributions associated with the individual features used by the ensemble.

We illustrate this idea in Fig. 1: In (a), we show an ensemble of decision stumps over three features $(x_1, x_2, x_3)$, aggregated to piecewise constant functions over one feature each (discussed in Section 3) and evaluated under the input randomization $\phi(\boldsymbol{x})$, here a Gaussian. We can compute the independent probability density functions of their outputs (PDFs) (shown in (b)) directly, by evaluating the (Gaussian) cumulative density function (CDF) over the constant regions. Aggregating the individual PDFs (discussed in Section 3), we can efficiently compute the exact PDF (c) and CDF (d) of the ensemble's output. To evaluate and certify the smoothed model, we can now simply look up the median prediction and success probability, respectively, in the CDF, without requiring sampling.

DRS combines $\ell_p$-norm certificates over numerical features, computed as described above, with an efficient worst-case analysis for $\ell_0$-perturbations of categorical features in order to, for the first time, provide joint certificates. To train models amenable to certification with DRS, we propose a robust MLE optimality criterion for training individual stumps and two boosting schemes targeting the certified robustness of the whole ensemble. We show empirically that DRS significantly improves on the state-of-the-art, increasing certified accuracies on established benchmarks up to *4-fold*.

**Main Contributions** Our key contributions are:

- DRS, a novel and efficient (De-)Randomized Smoothing approach for robustness certification, enabling joint deterministic certificates over numerical and categorical variables (Section 3).

- A novel MLE optimality criterion for training decision stumps robust under input randomization and two boosting approaches for certifiably robust stump ensembles (Section 4).

- An extensive empirical evaluation, demonstrating the effectiveness of our approach and establishing a new state-of-the-art in a wide range of settings (Section 5).

## 2   Background on Randomized Smoothing

For a given base model $F \colon \mathbb{R}^d \to [C]$, classifying inputs to one of $C \in \mathbb{Z}^{\geq 2}$ classes, Randomized Smoothing (RS) is a method to construct a classifier $G \colon \mathbb{R}^d \to [C]$ with robustness guarantees. For a randomization scheme $\phi \colon \mathbb{R}^d \to \mathbb{R}^d$, we define the success probability $p_y := \mathbb{P}_{\boldsymbol{x}' \sim \phi(\boldsymbol{x})}[F(\boldsymbol{x}') = y]$ and $G(\boldsymbol{x}) := \arg\max_{c \in [C]} p_y$. Depending on the choice of $\phi$, we obtain different certificates of the form:

**Theorem 2.1** (Adapted from Cohen et al. [24], Yang et al. [25]). *If* $\mathbb{P}(F(\phi(\boldsymbol{x})) = y) := p_y \geq \underline{p_y}$ *and* $\underline{p_y} > 0.5$, *then* $G(\boldsymbol{x} + \delta) = y$ *for all* $\delta$ *satisfying* $\|\delta\|_p < R$ *with* $R := \rho(\underline{p_y})$.

In particular, we present two instantiations that we utilize throughout this paper in Table 1, where $\Phi^{-1}$ is the inverse Gaussian CDF. Similar results, yielding other $\ell_p$-norm certificates, can be derived for a wide range of input randomization schemes [25, 26]. Note that, by using more information than just $p_y$, e.g., $p_c$ for the runner-up class $c$, tighter certificates can be obtained

Table 1: Randomized Smoothing guarantees.

| $\phi(\boldsymbol{x})$ | | $R := \rho(\underline{p_y})$ |
|---|---|---|
| $\ell_1$ | $\boldsymbol{x} + Unif([-\lambda, \lambda]^d)$ | $2\lambda(\underline{p_y} - \frac{1}{2})$ |
| $\ell_2$ | $\boldsymbol{x} + \mathcal{N}(\mathbf{0}, \sigma\mathbb{I})$ | $\sigma\Phi^{-1}(\underline{p_y})$ |

[24, 27]. Once $\underline{p_y}$ is computed, we can directly calculate the certifiable radius $R := \rho(p_y)$. For a broader overview of variants of Randomized Smoothing, please refer to Section 6.

For most choices of $F$ and $\phi$, the exact success probability $p_y$ can not be computed efficiently. Thus a lower bound $\underline{p_y}$ is estimated with confidence $1 - \alpha$ (typically $\alpha = 10^{-3}$) using Monte Carlo sampling and the Neyman-Pearson lemma [28]. Not only is this extremely computationally expensive, as typically $100\,000$ samples have to be evaluated per data point, but this also severely limits the maximum certifiable radius (see Fig. 6) and only yields probabilistic guarantees. Additionally, if the number of samples is not sufficient for the statistical test, the procedure will abstain from classifying.

In the following, we will show how considering a specific class of models $F$ allows us to compute the success probability $p_y$ exactly, overcoming these drawbacks, and thus invoke $\rho(p_y)$ to compute deterministic certificates over larger radii, orders of magnitude faster than RS.

## 3 (De-)Randomized Smoothing for Decision Stump Ensembles

Tree-based models such as decision stump ensembles often combine exceptional performance on tabular data [1] with good interpretability, making them ideal for many real-world high-stakes applications. Here, we propose a (De-)Randomized Smoothing approach, DRS, to equip them with deterministic robustness guarantees. For this, we first revisit decision stump ensembles and then show that their structure permits an exact evaluation under isotropic input randomization schemes, such as those discussed in Section 2. Finally, we propose joint certification over numerical and categorical variables, as many practical tabular datasets often contain both variable types.

**Stump Ensembles** We define a decision stump as $f_m(\boldsymbol{x}) = \gamma_{l,m} + (\gamma_{r,m} - \gamma_{l,m})\mathbb{1}_{x_{j_m} > v_m}$, with leaf predictions $\gamma_{l,m}, \gamma_{r,m} \in [0, 1]$, split position $v_m$, and split variable $j_m$. We construct unweighted ensembles, particularly suitable for Smoothing [29], of $M$ such stumps $\bar{f}_M \colon \mathbb{R}^d \mapsto [0, 1]$ as

$$\bar{f}_M(\boldsymbol{x}) := \frac{1}{M} \sum_{m=1}^{M} f_m(\boldsymbol{x}), \tag{1}$$

and treat them as a binary classifiers $\mathbb{1}_{\bar{f}_M(\boldsymbol{x}) > 0.5}$. While our approach is extensible to multi-class classification by replacing the scalar leaf predictions $\gamma$ with prediction-vectors, assigning a score per class, we focus on the binary case in this work.

**Smoothed Stump Ensemble** We now define a smoothed stump ensemble $\bar{g}_M$ along the lines of Randomized Smoothing as discussed in Section 2, by evaluating $\bar{f}_M$ not only on the original input $\boldsymbol{x}$ but rather on a whole distribution of $\boldsymbol{x}' \sim \phi(\boldsymbol{x})$:

$$\bar{g}_M(\boldsymbol{x}) := \mathbb{P}_{\boldsymbol{x}' \sim \phi(\boldsymbol{x})}[\bar{f}_M(\boldsymbol{x}') > 0.5].$$

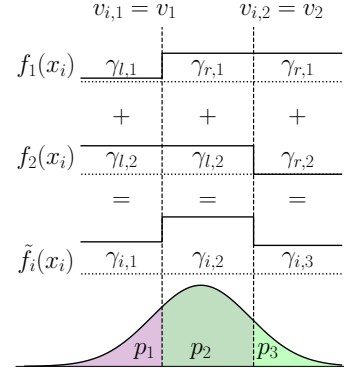

In this work, we consider randomization schemes $\phi(\boldsymbol{x})$ that are (i) isotropic, i.e., the dimensions of $\boldsymbol{x}' \sim \phi(\boldsymbol{x})$ are independently distributed, and (ii) permit an efficient computation of their marginal cumulative distribution functions (CDF). This includes a wide range of distributions, e.g., the Gaussian and Uniform distributions used in Table 1 and others commonly used for RS [25].

By denoting the model CDF as $\bar{\mathcal{F}}_{M,\boldsymbol{x}}(z) = \mathbb{P}_{\boldsymbol{x}' \sim \phi(\boldsymbol{x})}[\bar{f}(\boldsymbol{x}') \leq z]$, we can alternatively define $\bar{g}_M$ as $\bar{g}_M(\boldsymbol{x}) := 1 - \bar{\mathcal{F}}_{M,\boldsymbol{x}}(0.5)$, which will become useful later. For a label $y \in \{0, 1\}$ we obtain the success probability $p_y = |y - \bar{\mathcal{F}}_{M,\boldsymbol{x}}(0.5)|$ of predicting $y$ for a sample from $\phi(\boldsymbol{x})$.

Figure 2: A meta-stump constructed from two stumps.

---
**Algorithm 1** Stump Ensemble PDF computation via Dynamic Programming
---
$\quad$ **function** COMPUTEPDF($\{(\boldsymbol{\Gamma}, \boldsymbol{v})_i\}_{i=1}^d, \boldsymbol{x}, \phi$)
$\quad\quad$ pdf$[i][t] = 0$ for $t \in [M \cdot \Delta + 1], i \in [d]$
$\quad\quad$ pdf$[0][0] = 1$ $\qquad\qquad\qquad$ ▷ For 0 stumps all probability mass is on 0
$\quad\quad$ **for** $i = 1$ to $d$ **do**
$\quad\quad\quad$ **for** $j = 1$ to $M_i$ **do**
$\quad\quad\quad\quad$ **for** $t = 0$ to $M \cdot \Delta + 1 - \Gamma_{i,j}$ **do**
$\quad\quad\quad\quad\quad$ pdf$[i][t + \Gamma_{i,j}]$ = pdf$[i][t + \Gamma_{i,j}]$ + pdf$[i-1][t] \cdot \mathbb{P}_{x' \sim \phi(\boldsymbol{x})}[v_{i,j-1} < x'_i \leq v_{i,j}]$
$\quad\quad$ **return** pdf
---

**Meta-Stumps** To evaluate $p_y$ exactly as illustrated in Fig. 1, we group the stumps constituting an ensemble by their split variable $j_m$ to obtain one *meta-stump* $\tilde{f}_i$ per feature $i$. The key idea is that outputs of these meta-stumps are now independently distributed under isotropic input randomization (illustrated in Fig. 1 (b)), allowing us to aggregate them efficiently later on.

We showcase this in Fig. 2, where two stumps ($f_1$ and $f_2$) are combined into the meta-stump $\tilde{f}_i$. Formally, we have

$$\tilde{f}_i(\boldsymbol{x}) := \sum_{m \in \mathcal{I}_i} f_m(\boldsymbol{x}), \qquad \mathcal{I}_i := \{m \in [M] \mid j_m = i\}, \qquad (2)$$

define $M_i = |\mathcal{I}_i|$ and rewrite our ensemble as $\bar{f}_M(\boldsymbol{x}) := \frac{1}{M} \sum_{i=1}^d \tilde{f}_i(x_i)$. Every meta-stump can be represented by its split positions $v_{i,j}$, sorted such that $v_{i,j} \leq v_{i,j+1}$, and its predictions $\gamma_{i,j} = \sum_{t=1}^{j-1} \gamma_{r,t} + \sum_{t=j}^{|\mathcal{I}_i|} \gamma_{l,m}$ on each of the resulting $|\mathcal{I}_i| + 1$ regions, written as $(\boldsymbol{\gamma}, \boldsymbol{v})_i$.

**CDF Computation** Now we leverage the independence of our meta-stumps' output distributions under an isotropic input randomization scheme $\phi$ to compute the PDF of their ensemble efficiently via dynamic programming (DP) (illustrated in Fig. 1 (c) and explained below). Given its PDF, we can trivially compute the ensemble's CDF $\bar{\mathcal{F}}_{M,\boldsymbol{x}}$, allowing us to evaluate the smoothed model exactly (illustrated in Fig. 1 (d)). This efficient CDF computation constitutes the core of DRS.

In more detail, we observe that the PDF of a stump ensemble is the convex sum of exponentially many ($\mathcal{O}((\max_i \mathcal{I}_i)^d)$) Dirac-delta distributions. To avoid this exponential blow-up, we discretize all leaf predictions $\gamma$ to a grid of $\Delta$ values (typically $\Delta = 100$), when constructing the smoothed model $\bar{g}_M$. For each $\gamma_{i,j}$, we define a corresponding $\Gamma_{i,j} \in \{0, \ldots, M_i \cdot \Delta\}$ such that $\gamma_{i,j} = \frac{\Gamma_{i,j}}{\Delta}$. Now, we construct a DP-table, where every entry pdf[i][t] corresponds to the weight of the Dirac-delta associated with an output of $\frac{t}{\Delta M}$ after considering the first $i$ meta-stumps (in any arbitrary but fixed order). We show the PDF computation in Algorithm 1 and provide an intuition below. We initialize pdf[0][*] by allocating all probability mass to t = 0 (pdf[0][0]=1). Now, we compute pdf[i][*] from pdf[i-1][*] by accounting for the effect of the $i^{\text{th}}$ meta-stump as follows: The weight of the Dirac-delta at t after considering i meta-stumps is exactly the sum over the weights of the Dirac-deltas at t-$\Gamma_{i,j}$ after i-1 meta-stumps, weighted with the probability $p_{i,j} := \mathbb{P}_{x' \sim \phi(\boldsymbol{x})}[v_{i,j-1} < x'_i \leq v_{i,j}]$ of the $i^{\text{th}}$ meta stump predicting $\Gamma_{i,j}$. We compute $p_{i,j}$ as the probability of the randomized $x'_i$ lying between $v_{i,j-1}$ and $v_{i,j}$ (padded with $-\infty$ and $\infty$ on the left and right, respectively), as illustrated in Fig. 2. After termination, the last line of the DP-table pdf[d][*] contains the full PDF (see Fig. 1(c)). Formally we summarize this in the theorem below, delaying a formal proof to App. A.1:

**Theorem 3.1.** *For* $z \in [0, 1]$, $\bar{\mathcal{F}}_{M,\boldsymbol{x}}(z) = \sum_{t=0}^{\lfloor zM\Delta \rfloor} pdf[d][t]$ *describes the exact CDF and thus success probability* $p_y = \mathbb{P}_{\boldsymbol{x}' \sim \phi(\boldsymbol{x})}[\bar{f}_M(\boldsymbol{x}') = y] = |y - \bar{\mathcal{F}}_{M,\boldsymbol{x}}(0.5)|$ *for* $y \in \{0, 1\}$.

Note that the presented algorithm is slightly simplified, and we actually only have to track the range of non-zero entries of one row of the DP-table. This allows us to compute the full PDF and thus certificates for smoothed stump ensembles very efficiently, e.g., taking only around 1.2 s total for the MNIST 2 vs. 6 task (around 2.000 data points and over 500 stumps).

**Certification** Recall from Section 2 that, given the success probability $p_y$, robustness certification for $\ell_p$-norm bounded perturbations reduces to computing the maximal certifiable robustness radius $R = \rho(p_y)$. For all popular $\ell_p$-norms, $\rho$ (and its inverse $\rho^{-1}$; used shortly) can be either evaluated symbolically [24, 25] or precomputed efficiently [30, 31], such that the core challenge of certification

becomes computing (a lower bound to) $p_y$, which we solve efficiently via Theorem 3.1. Alternatively, for a given target radius $r$, we need to check whether $p_y \geq \rho^{-1}(r)$ by equivalently calculating

$$\bar{g}_{M,r}(\boldsymbol{x}) = \bar{\mathcal{F}}_{M,\boldsymbol{x}}^{-1}(z) \qquad z = \begin{cases} 1 - \rho^{-1}(r) & \text{if } y = 1 \\ \rho^{-1}(r) & \text{if } y = 0 \end{cases}, \qquad (3)$$

and checking $\bar{g}_{M,r}(\boldsymbol{x}) > 0.5$. This corresponds to asserting that class $y$ is predicted at least $z$ of the time. Here, the inverse CDF $\bar{\mathcal{F}}_{M,\boldsymbol{x}}^{-1}(z)$ can be efficiently evaluated using the step-wise $\bar{\mathcal{F}}_{M,\boldsymbol{x}}$ computed via Theorem 3.1. We will see in Section 4 that this view is useful when training stump ensembles for certifiability. Finally, we want to highlight that this approach can be used with all common randomization schemes yielding certificates for different $\ell_p$-norm bounded adversaries.

**Categorical Variables & Joint Certificates** For practical applications, it is essential to handle both numerical and categorical features jointly. To consider a categorical feature $x_i \in \{1, \dots, d_i\}$ in our stump ensemble, we construct a $d_i$-ary stump $\tilde{f}_i \colon [d_i] \to [0,1]$ returning a value $\gamma_{i,j}$ corresponding to each of the $d_i$ categorical values and treated as a meta-stump with $M_i = 1$ for normalization.

To provide certificates in this setting, we propose a novel scheme combining an arbitrary $\ell_p$-norm certificate of radius $r_p$ over all numerical features, computed as discussed above, with an $\ell_0$ certificate of radius $r_0$ over all categorical features $\mathcal{C}$, computed using an approach adapted from Wang et al. [23]. Conceptually, we compute the worst-case effect of every individual categorical variable independently, greedily aggregate these worst-case effects, and account for them in our ensemble's CDF.

Given a meta-stump's prediction on a concrete sample $q_i = \tilde{f}_i(x_i)$ as well as its maximal and minimal output $u_i$ and $l_i$, respectively, we compute the maximum and minimum perturbation effect to $\overline{\delta}_i = \frac{u_i - q_i}{M}$ and $\underline{\delta}_i = \frac{l_i - q_i}{M}$, respectively. Given the set of categorical features $\mathcal{C}$, we can compute the worst-case effect when perturbing at most $r_0$ samples as

$$\overline{\delta}_{r_0} = \max_{\mathcal{R}} \sum_{i \in \mathcal{R}} \overline{\delta}_i, \quad s.t. |\mathcal{R}| \leq r_0, \mathcal{R} \subseteq \mathcal{C}$$

by greedily picking the $r_0$ largest $\overline{\delta}_i$. For $\underline{\delta}_{r_0}$ we proceed analogously. Shifting the CDF, computed as above, by $\overline{\delta}$ and $\underline{\delta}$ for samples with labels $y = 0$ and $y = 1$, respectively, before computing the success probability $p_y$, allows us to account for the worst-case categorical perturbations exactly. We illustrate this for a sample with $y = 0$ in Fig. 3, where we show the CDFs obtained by all possible perturbations of at most $r_0$ categorical variables, bounded to the right by those obtained by shifting the original by $\overline{\delta}_{r_0}$. Note that here no smoothing over the categorical variables is done or required, making inference trivial.

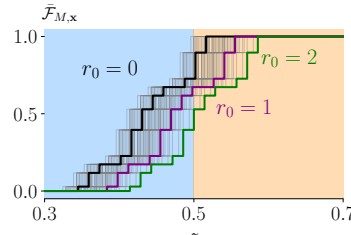

Figure 3: CDF shifted by the effect of categorical feature perturbations.

## 4 Training for and with (De-)Randomized Smoothing

To obtain large certified radii via smoothing, the base model has to be robust to the chosen randomization scheme. To train robust decision stump ensembles, we propose a robust MLE optimality criterion for individual stumps (Section 4.1) and two boosting schemes for whole ensembles (Section 4.2).

### 4.1 Independently MLE-Optimal Stumps

To train an individual stump $f_m(\boldsymbol{x}) = \gamma_{l,m} + (\gamma_{r,m} - \gamma_{l,m})\mathbb{1}_{x_{j_m} > v_m}$, its split feature $j_m$, split position $v_m$, and leaf predictions $\gamma_{l,m}, \gamma_{r,m}$ have to be determined. We choose them in an MLE-optimal fashion with respect to the randomization scheme $\phi$, starting with $v_m$, as follows: We consider the probabilities $p_{l,i}(v_m) = \mathbb{P}_{\boldsymbol{x}' \sim \phi(\boldsymbol{x}_i)}[x'_{j_m} \leq v_m]$ and $p_{r,i} = 1 - p_{l,i}(v_m)$ of $\boldsymbol{x}'_i$ lying to the left or the right of $v_m$, respectively, under the input randomization scheme $\phi$. To avoid clutter, we drop the explicit dependence on $v_m$ in the following. For an i.i.d. dataset with $n$ samples $(\boldsymbol{x}_i, y_i) \sim (\mathcal{X}, \mathcal{Y})$, we define the probabilities $p_j^y = \frac{1}{n} \sum_{\{i | y_i = y\}} p_{j,i}$ of picking the $j \in \{l, r\}$ leaf, conditioned on the target label, and $p_j = p_j^0 + p_j^1$ as their sum to compute the entropy impurity $H_{\text{entropy}}$ [32] as

$$H_{\text{entropy}} = - \sum_{j \in \{l,r\}} p_j \sum_{y \in \{0,1\}} \frac{p_j^y}{p_j} \log\left(\frac{p_j^y}{p_j}\right).$$

We then choose the $v_m$ approximately minimizing $H_{\text{entropy}}$ via line-search. After fixing $v_m$ this way, we compute the MLE-optimal leaf predictions $\gamma_l^{\phi,\text{MLE}}$ and $\gamma_r^{\phi,\text{MLE}}$ as:

$$
\begin{aligned}
\gamma_l^{\phi\text{MLE}}, \gamma_r^{\phi\text{MLE}} &= \arg\max_{\gamma_l, \gamma_r} \mathbb{P}[\mathcal{Y} \mid \phi(\mathcal{X}), f_m] = \arg\max_{\gamma_l, \gamma_r} \sum_{i=1}^{n} \mathbb{E}_{\boldsymbol{x}' \sim \phi(\boldsymbol{x}_i)} \left[\log \mathbb{P}[y_i \mid \boldsymbol{x}', f_m]\right] \\
&= \arg\max_{\gamma_l, \gamma_r} \sum_{i \in \{i|y_i=0\}}^{n} p_{l,i} \log(1 - \gamma_l) + p_{r,i} \log(1 - \gamma_r) \\
&\quad + \sum_{i \in \{i|y_i=1\}}^{n} p_{l,i} \log(\gamma_l) + p_{r,i} \log(\gamma_r) \\
&= \arg\max_{\gamma_l, \gamma_r} p_l^0 \log(1 - \gamma_l) + p_r^0 \log(1 - \gamma_r) + p_l^1 \log(\gamma_l) + p_r^1 \log(\gamma_r),
\end{aligned}
$$

where the second line is obtained by splitting the sum over samples by class and explicitly computing the expectation. We solve the maximization problem by setting the first derivatives $\frac{\partial}{\partial \gamma_l}$ and $\frac{\partial}{\partial \gamma_r}$ of our optimization objective to zero and checking its Hessian to confirm that

$$
\gamma_l^{\phi\text{MLE}} = \frac{p_l^1}{p_l^1 + p_l^0} \qquad \gamma_r^{\phi\text{MLE}} = \frac{p_r^1}{p_r^1 + p_r^0} \tag{4}
$$

are indeed maxima. We show in App. A.2 that $\gamma_l^{\phi,\text{MLE}}, \gamma_l^{\phi,\text{MLE}}$, and $v_m$ are even jointly MLE-optimal, when $v_m$ is chosen as the exact instead of an approximate minimizer of the entropy impurity.

**Ensembling** To train an ensemble of independently MLE-optimal decision stumps, we sequentially train one stump for every feature $j_m \in [d]$ and construct an ensemble with equal weights, rejecting stumps with an entropy impurity $H_{\text{entropy}}$ above a predetermined threshold.

## 4.2 Boosting Stump Ensembles for Certifiable Robustness

Decision stumps trained this way maximize the expected likelihood under the chosen randomization scheme. Assuming (due to the law of large numbers) a Gaussian output distribution, this corresponds to optimizing for the median output, which determines the clean prediction. However, certified correctness at a given radius $r$ is determined by the prediction $y'(\boldsymbol{x}, r) = \bar{\mathcal{F}}_{m-1,\boldsymbol{x}}^{-1}(z(r))$ at the $z(r) := |y - \rho^{-1}(r)|$ percentile of the output distribution. Where we call $y'$ the *certifiable prediction*, as certification is now equivalent to checking $y = \mathbb{1}_{y'(\boldsymbol{x},r)>0.5}$ (Eq. (3)). This difference is illustrated in Fig. 4, where the clean prediction is correct (class 1) while the certifiable prediction is incorrect. To align our training objective better with certified accuracy, we propose two novel boosting schemes along the lines of the popular TREEBOOST [33] and ADABOOST [34].

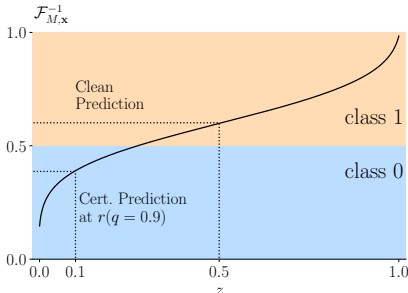

Figure 4: Inverse CDF $\bar{\mathcal{F}}_M^{-1}$

**Gradient Boosting for Certifiable Robustness** The key idea of gradient boosting is to compute the gradient of a loss function with respect to an ensemble's outputs and then add a model to the ensemble that makes a prediction along this gradient direction. Implementing this idea, we adapt TREEBOOST [33] to propose ROBTREEBOOST: At a high level, we add stumps to the ensemble, which aim to predict the residual between the target label and the current certifiable prediction. Concretely, to add the $m^{\text{th}}$ stump to our ensemble, we begin by computing the current ensemble's certifiable predictions $y'(r)$ at a target radius $r$ and then defining the pseudo labels $\tilde{y} = y - y'(r)$ as the residual between the target labels $y$ and the certifiable predictions $y'(r)$. This yields a regression problem, which we tackle by choosing a feature $j_m$ and split threshold $v_m$ (approximately) minimizing MSE impurity under input randomization before computing $\gamma_{l,m}$ and $\gamma_{r,m}$ as approximate minimizers of the cross-entropy loss over the whole ensemble. Please see App. A.3 for a more detailed discussion of ROBTREEBOOST.

Table 2: Natural accuracy (NAC) [%] and certified accuracy (CA) [%] with respect to $\ell_1$- and $\ell_2$-norm bounded perturbations. Results for Wang et al. [23] as reported by them. Larger is better.

| Perturbation | Dataset | Radius $r$ | Standard Training | Wang et al. [23] | | Ours (Independent) | | Ours (Boosting) | |
|---|---|---|---|---|---|---|---|---|---|
| | | | NAC | NAC | CA | NAC | CA | NAC | CA |
| $\ell_1$-norm | BREASTCANCER | 1.0 | 99.3 | 98.5 | 64.2 | **100.0** | 81.0 | **100.0** | **83.9** |
| | DIABETES | 0.05 | 74.7 | 72.7 | 68.2 | 76.0 | 69.5 | **77.9** | **72.1** |
| | FMNIST-SHOES | 0.5 | **95.0** | 87.6 | 67.8 | 85.8 | 83.3 | 87.2 | **84.2** |
| | MNIST 1 vs. 5 | 1.0 | 99.1 | 95.5 | 83.8 | 96.6 | 94.1 | **99.3** | **98.1** |
| | MNIST 2 vs. 6 | 1.0 | 96.0 | 92.3 | 66.5 | 96.3 | 93.9 | **96.6** | **94.1** |
| $\ell_2$-norm | BREASTCANCER | 0.7 | 99.3 | 91.2 | 60.6 | **100.0** | 75.2 | **100.0** | **82.5** |
| | DIABETES | 0.05 | 74.7 | - | - | 77.3 | 68.2 | **79.9** | **71.4** |
| | FMNIST-SHOES | 0.4 | **95.0** | 75.5 | 51.5 | 86.8 | 81.2 | 91.0 | **84.5** |
| | MNIST 1 vs. 5 | 0.8 | 99.1 | 95.6 | 63.4 | 95.8 | 91.6 | **99.2** | **96.3** |
| | MNIST 2 vs. 6 | 0.8 | 96.0 | 86.3 | 23.0 | **96.3** | 89.6 | **96.3** | **89.6** |

**Adaptive Boosting for Certifiable Robustness** The key idea of adaptive boosting is to build an ensemble by iteratively training models, weighted based on their error rate, while adapting sample weights based on whether they are classified correctly. We build on ADABOOST [34] to propose ROBADABOOST: We construct an ensemble of $K$ stump ensembles via hard voting, where every ensemble is weighted based on its certifiable accuracy. To train a new ensemble, we increase the weights of all samples that are currently not classified certifiably correctly at a given radius $r$. We choose stump ensembles instead of individual stumps as base classifiers because single stumps often can not reach the success probabilities under input randomization required for certification. To compute the certifiable radius for such an ensemble $\bar{F}_K$, we compute the certifiable radii $R^k$ of the individual stump ensembles $\bar{f}_M^k$, sort them in decreasing order such that $R^k \geq R^{k+1}$ and obtain the largest radius $R^k$ such that the weights of the first $k$ ensembles sum up to more than half of the total weights. Please see App. A.4 for a more detailed discussion of ROBADABOOST.

## 5 Experimental Evaluation

In this section, we empirically demonstrate the effectiveness of DRS in a wide range of settings. We show that DRS significantly outperforms the current state-of-the-art for certifying tree-based models on established benchmarks, using only numerical features (Section 5.1), before highlighting its novel ability to obtain joint certificates on a set of new benchmarks (Section 5.2). Finally, we perform an ablation study, investigating the effect of DRS's key components (Section 5.3).

**Experimental Setup** We implement our approach in PyTorch [35] and evaluate it on Intel Xeon Gold 6242 CPUs and an NVIDIA RTX 2080Ti. We compare to prior work on the DIABETES [36], BREASTCANCER [37], FMNIST-SHOES [38], MNIST 1 vs. 5 [39], and MNIST 2 vs. 6 [39] datasets and are the first to provide joint certificates of categorical and numerical features, demonstrated on the ADULT [37] and CREDIT [37] datasets. For a more detailed description of the experimental setup, please refer to App. B.

### 5.1 Certification for Numerical Features

In Table 2, we compare the certified accuracies obtained via DRS on ensembles of independently MLE optimal stumps (Independent) or boosted stump ensembles (Boosting) to the current state-of-the-art, Wang et al. [23], and standard training [40] using established benchmarks [23].

**Independently MLE Optimal Stumps** We first consider stump ensembles trained without boosting as described in Section 4.1 and observe that DRS obtains higher certified accuracies in all settings and higher natural accuracies in most. For example, on MNIST 2 vs. 6, we increase the certified accuracy at an $\ell_2$ radius of $r_2 = 0.8$ from 23.0% to 89.6%, almost quadrupling it compared to Wang et al. [23], while also improving natural accuracy from 86.3% to 96.3%.

**Boosting for Certified Accuracy** Leveraging the boosting techniques introduced in Section 4.2, ROBTREEBOOST for BREASTCANCER and DIABETES and ROBADABOOST for FMNIST-SHOES, MNIST 1 vs. 5, and MNIST 2 vs. 6, we increase certifiable and natural accuracies even further in most settings. For example, compared to our independently trained stump ensemble, we improve the certified accuracy for MNIST 1 vs. 5 at an $\ell_1$-radius of $r_1 = 1.0$ from 94.1% to 98.1% and for BREASTCANCER at an $\ell_2$-radius of $r_2 = 0.7$ from 75.2% to 82.5%.

Table 3: Balanced certified accuracy (BCA) [%] under joint $\ell_0$- and $\ell_2$-perturbations of categorical and numerical features, respectively, depending on whether model uses categorical and/or numerical features. The balanced natural accuracy is the BCA at radius $r = 0.0$. Larger is better.

| Dataset | Categorical Features | $\ell_0$ Radius $r_0$ | BCA without Numerical Features | BCA with Numerical Features at $\ell_2$ Radius $r_2$ | | | | | | |
|---|---|---|---|---|---|---|---|---|---|---|
| | | | | 0.00 | 0.25 | 0.50 | 0.75 | 1.00 | 1.25 | 1.50 |
| ADULT | no | - | - | 74.9 | 65.7 | 42.4 | 27.4 | 14.5 | 8.9 | 5.1 |
| | yes | 0 | 76.6 | 77.5 | 73.9 | 68.1 | 63.3 | 48.7 | 40.7 | 35.2 |
| | | 1 | 57.4 | 66.0 | 61.7 | 53.9 | 47.4 | 34.3 | 26.6 | 21.8 |
| | | 2 | 33.5 | 51.4 | 46.2 | 37.5 | 29.3 | 21.5 | 17.1 | 13.4 |
| | | 3 | 8.9 | 36.7 | 31.4 | 24.1 | 15.4 | 10.3 | 8.1 | 5.7 |
| CREDIT | no | - | - | 56.1 | 44.5 | 33.3 | 17.7 | 9.7 | 7.2 | 5.0 |
| | yes | 0 | 70.7 | 74.1 | 70.3 | 67.3 | 59.7 | 57.1 | 54.9 | 53.4 |
| | | 1 | 48.2 | 52.7 | 47.7 | 41.7 | 38.3 | 37.1 | 35.1 | 34.7 |
| | | 2 | 26.4 | 29.3 | 26.0 | 23.8 | 19.2 | 16.8 | 13.5 | 13.0 |
| | | 3 | 7.8 | 13.6 | 10.3 | 7.8 | 4.9 | 4.4 | 3.9 | 3.4 |

## 5.2 Joint Certificates for Categorical and Numerical Features

In Table 3, we compare models using only numerical, only categorical, or both types of features with regards to their balanced certified accuracy (BCA) (accounting for class frequency) at different combinations of $\ell_2$- and $\ell_0$-radii for numerical and categorical features, respectively. We observe that models using both categorical and numerical features perform notably better on clean data, highlighting the importance of utilizing and thus also certifying them in combination. Moreover, categorical features make the model significantly more robust to $\ell_2$ perturbations, e.g., at $\ell_2$-radii $\geq 0.75$, they improve certified accuracies, even when 2 categorical features (of only 8 and 7 for ADULT and CREDIT, respectively)

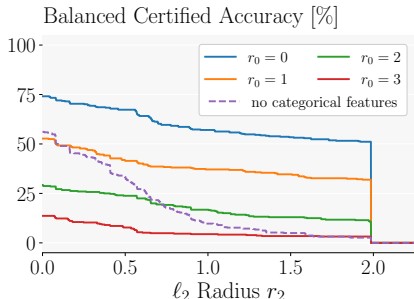

Figure 5: Effect of $\ell_0$-perturbations on $\ell_2$-robustness for CREDIT.

are adversarially perturbed. We visualize this in Fig. 5, showing BCA over $\ell_2$-perturbation radius and confirm that the model utilizing only numerical features (dotted line) loses accuracy much quicker with perturbation magnitude than the model leveraging categorical variables (solid lines). As we are the first to tackle this setting, we do not compare to other methods but provide more detailed experiments in App. C.1.

## 5.3 Ablation Study

We first illustrate the effectiveness of our derandomization approach, before demonstrating the benefit of training with our MLE optimality criterion and investigating the effect of the noise level on DRS.

**(De-)Randomized vs Randomized Smoothing** In Fig. 6, we compare DRS, (dotted line) and sampling-based RS (solid lines), w.r.t. certified accuracy over $\ell_2$ radii. We observe that the sampling-based estimation of the success probability in RS significantly limits the obtained certifiable radii. While this effect is particularly pronounced for small sample counts $n$, increasing the maximum certifiable radius, visible as the sudden drop in certifiable accuracy, requires an exponentially increasing number of samples, making the certification of large radii intractable. DRS, in contrast, can compute exact success probabilities and thus deterministic guarantees for much larger radii, yielding a 33.1% increase in ACR compared to using $n = 100\,000$

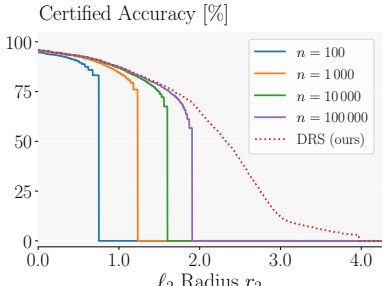

Figure 6: DRS vs. RS with various sample counts $n$ on MNIST 1 vs. 5.

samples. Additionally, DRS is multiple orders of magnitude faster than RS, here, only requiring approximately $6.45 \cdot 10^{-4}$ s per sample. For more extensive experiments, please refer to App. C.2.

**MLE Optimality Criterion** In Table 4, we evaluate our robust MLE optimality criterion (MLE) by comparing it to the standard entropy criterion applied to samples drawn from the input randomization scheme (Sampling) or the clean data (Default). We observe that the ensemble trained on the clean data (Default) suffers from a mode collapse when evaluated under noise. In contrast, both approaches considering the input randomization perform much better, with our robust MLE approach outperforming sampling by a significant margin, especially at large radii. For more extensive experiments, please refer to App. C.3.

**Effect of Noise Level** In Fig. 7, we compare the certified accuracy over $\ell_1$-radii for a range of different noise magnitudes $\lambda$ and ensembles of independently MLE optimal stumps. We observe that at large perturbation magnitudes, we obtain stumps that 'think outside the (hyper-)box', i.e., choose splits outside of the original data range, making their ensembles exceptionally robust, even at large radii. In particular, we obtain a certifiable accuracy of 87.3% at radius $r_1 = 4.0$, while the state-of-the-art achieves only 83.8% at $r_1 = 1.0$ [23]. We provide more experiments for varying noise magnitudes in App. C.4.

Table 4: Comparison of training with the exact distribution (MLE), randomly perturbed data (Sampling), or clean data (Default) on BREAST-CANCER for $\sigma = 1$.

| Method | ACR | Certified Accuracy [%] at Radius $r$ | | | |
| --- | --- | --- | --- | --- | --- |
| | | 0.0 | 0.25 | 0.5 | 0.75 |
| MLE (Ours) | **0.675** | **100.0** | **97.1** | **86.1** | **30.7** |
| Sampling | 0.567 | 99.3 | 95.6 | 75.2 | 8.8 |
| Default | 0.356 | 26.3 | 25.5 | 25.5 | 25.5 |

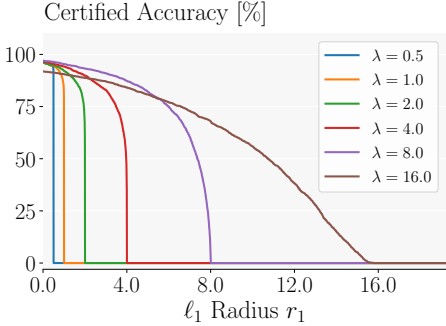

Figure 7: Comparing DRS for various noise levels $\lambda$ on MNIST 1 VS. 5.

# 6 Related Work

**(De-)Randomized Smoothing** Probabilistic certification methods [41, 42, 24] are a popular approach for obtaining robustness certificates for a wide range of tasks [31, 43–46], threat models [25–27, 30, 31, 43, 47–53], and robustness-accuracy trade-offs [54]. These methods follow the general blueprint discussed in Section 2 and consider arbitrary base classifiers, though specially trained [55–57]. While recent work [29, 58] has found ensembles to be particularly suitable base classifiers, they use neural networks and can thus, in contrast to our work, not leverage their structure. Specifically designed randomization schemes [51, 53] enable efficient enumeration and thus a deterministic certificate for, e.g., patch attacks or $\ell_1$-norm perturbations. In contrast to these approaches, we permit arbitrary isotropic continuous randomization schemes, allowing us to leverage comprehensive results on RS to obtain robustness guarantees against a wide range of $\ell_p$-norm bounded adversaries [25].

**Certification and Training of Tree-Based Models** In the setting of $\ell_\infty$ robustness, where every feature can be perturbed independently, various methods have been proposed to train [4, 22, 59–61] and certify [22, 62–64] robust decision trees and stumps. However, $\ell_\infty$ robust models are still vulnerable to other $\ell_p$ norm perturbations [65, 66], which cover many realistic perturbations better and are the focus of this work. There, the admissible perturbation of one feature depends on the perturbations of all others, making the above approaches leveraging their independence not applicable.

On the other hand, Kantchelian et al. [67] discuss complete robustness certification of tree ensembles in the $\ell_p$-norm setting via MILP. However, this approach is intractable in most settings due to its Co-NP-complete complexity. Wang et al. [23] propose an efficient but incomplete DP-based certification algorithm for stump ensembles based on over-approximating the maximum perturbation effect in the $\ell_p$-norm setting. While similarly fast as our approach, we show empirically in Section 5 that DRS obtains significantly stronger certificates. Wang et al. [23] further introduce an incomplete certification algorithm for tree ensembles, which is based on computing the distance between the pre-image of all trees' leaves and the original sample. As they report significantly worse results using this approach than with stump ensembles, we omit a detailed comparison.

# 7    Limitations and Societal Impact

**Limitations**  While able to handle arbitrary stump ensembles, and being extensible to arbitrary decision trees (see App. D), DRS can not handle arbitrary ensembles of decision trees. However, as these have been shown to be significantly more sensitive to $\ell_p$-norm perturbations than stump ensembles [23], we believe this limitation to be of little practical relevance. Further, like all Smoothing-based approaches, we construct a smoothed model from a base classifier and only obtain robustness guarantees for the former. In contrast to standard Randomized Smoothing approaches, we can, however, evaluate the smoothed model exactly and efficiently.

**Societal Impact**  As our contributions improve certified accuracy and certification radii while retaining high natural accuracy, they could help make real-world AI systems more robust and thus generally amplify both any positive or negative societal effects. Further, while we achieve state-of-the-art results, these may not be sufficient to guarantee robustness in real-world deployment and could give practitioners a false sense of security, leading to them relying more on our models than is justified.

# 8    Conclusion

We propose DRS, a (De-)Randomized Smoothing approach to robustness certification, enabling joint deterministic certificates over numerical and categorical variables for decision stump ensembles by leveraging their structure to compute their exact output distributions for a given input randomization scheme. The key insight enabling this is that this output distribution can be efficiently computed by aggregating independent distributions associated with the individual features used by the ensemble. We additionally propose a robust MLE optimality criterion for training individual decision stumps and two boosting schemes improving an ensemble's certifiable accuracy. Empirically, we demonstrate that DRS significantly outperforms the state-of-the-art for tree-based models in a wide range of settings, obtaining up to 4-fold improvements in certifiable accuracy.

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
