# A  Additional Theory

In this section, we provide additional theoretical results, omitted in the main paper due to space constraints. Concretely, in App. A.1 we provide a proof for Theorem 3.1 on the correctness of our PDF computation. In App. A.2, we show that $\gamma_l$, $\gamma_r$ and $v_m$, computed as outlined in Section 4, are indeed jointly MLE-optimal. Finally, we provide more details on ROBTREEBOOST and ROBADABOOST in App. A.3 and App. A.4, respectively.

## A.1  PDF Computation

Here, we provide a proof for Theorem 3.1 on the correctness of our efficient PDF-computation, restated below for convenience.

**Theorem 3.1.** *For $z \in [0, 1]$, $\bar{\mathcal{F}}_{M,\boldsymbol{x}}(z) = \sum_{t=0}^{\lfloor zM\Delta \rfloor} \mathsf{pdf}[d][t]$ describes the exact CDF and thus success probability $p_y = \mathbb{P}_{\boldsymbol{x}' \sim \phi(\boldsymbol{x})}[\bar{f}_M(\boldsymbol{x}') = y] = |y - \bar{\mathcal{F}}_{M,\boldsymbol{x}}(0.5)|$ for $y \in \{0, 1\}$.*

*Proof.* Let the random variable $\Gamma^{(i)}$ be the prediction of the $i$-th meta-stump, then we have by definition of the meta-stump $\mathbb{P}[\Gamma^{(i)} = \Gamma_{i,j}] = \mathbb{P}_{x'_i \sim \phi(\boldsymbol{x})}[v_{i,j-1} < x'_i \leq v_{i,j}]$ (see Section 3). Note that, for presentational simplicity, we assume $\Gamma_{i,j} \neq \Gamma_{i,k}, \forall k \neq j$. Now, we first show by induction that $\mathsf{pdf}[i]$ computes the exact PDF of $\sum_{l=1}^{i} \Gamma^{(l)}$ (Lemma 1), before showing how the CDF of the meta-stump ensemble follows.

**Lemma 1.** *Algorithm 1 computes $\mathsf{pdf}[i][t] = \mathbb{P}\left[\sum_{l=1}^{i} \Gamma^{(l)} = t\right]$.*

*Proof.* We proceed by induction over $i$. In the base case, for $i = 0$, we directly have $\mathsf{pdf}[0][0] = 1.0$ and $\mathsf{pdf}[0][t] = 0.0$ for $t > 0$ by construction. Now the induction assumption is that $\mathsf{pdf}[i-1][t] = \mathbb{P}\left[\sum_{l=1}^{i-1} \Gamma^{(l)} = t\right]$ for an arbitrary $i \leq d$ and all corresponding $t$. To compute the $\mathsf{pdf}[i][t]$, we now have:

$$
\begin{aligned}
\mathsf{pdf}[i][t] &= \sum_{j=1}^{M_i} \mathsf{pdf}[i-1][t - \Gamma_{i,j}] \cdot \mathbb{P}_{x'_i \sim \phi(\boldsymbol{x})}[v_{i,j-1} < x'_i \leq v_{i,j}] \\
&= \sum_{j=1}^{M_i} \mathbb{P}\left[\left(\sum_{l=1}^{i-1} \Gamma^{(l)}\right) = t - \Gamma_{i,j}\right] \cdot \mathbb{P}_{x'_i \sim \phi(\boldsymbol{x})}[v_{i,j-1} < x'_i \leq v_{i,j}] \\
&= \sum_{j=1}^{M_i} \mathbb{P}\left[\left(\sum_{l=1}^{i} \Gamma^{(l)}\right) = t \,\middle|\, \Gamma^{(i)} = \Gamma_{i,j}\right] \cdot \mathbb{P}\left[\Gamma^{(i)} = \Gamma_{i,j}\right] \\
&= \mathbb{P}\left[\left(\sum_{l=1}^{i} \Gamma^{(l)}\right) = t\right]
\end{aligned}
$$

where we first use the definition $\mathsf{pdf}[i][t]$ according to Algorithm 1, followed by induction assumption, the independency of different meta-stumps and the the law of total probability over $j$. $\qquad\square$

Now, we show how Theorem 3.1 directly follows from Lemma 1. Recall that $\gamma_{i,j} = \frac{\Gamma_{i,j}}{\Delta}$, where $\Delta$ is the number of discretization steps. Similarly to $\Gamma^{(i)}$, let $\gamma^{(i)}$ be the random variable describing the prediction of the $i$-th meta-stump. Using Lemma 1, we obtain

$$
\begin{aligned}
\bar{\mathcal{F}}_{M,\boldsymbol{x}}(z) &= \sum_{t=0}^{\lfloor zM\Delta \rfloor} \mathsf{pdf}[d][t] \\
&= \sum_{t=0}^{\lfloor zM\Delta \rfloor} \mathbb{P}\left[\sum_{i=1}^{d} \Gamma^{(i)} = t\right]
\end{aligned}
$$

$$= \mathbb{P}\left[\sum_{i=1}^{d} \Gamma^{(i)} \leq \lfloor zM\Delta \rfloor\right]$$

$$= \mathbb{P}\left[\sum_{i=1}^{d} \frac{\gamma^{(i)}}{M} M\Delta \leq \lfloor zM\Delta \rfloor\right]$$

$$= \mathbb{P}\left[\sum_{i=1}^{d} \frac{\gamma^{(i)}}{M} \leq z\right]$$

$$= \mathbb{P}\left[\bar{f}_M(\boldsymbol{x}) \leq z\right].$$

Where the second to last step follows from the discretization of the leaf predictions leading to a piece-wise constant CDF. $\qquad\square$

## A.2 MLE-Optimal Stumps

In this section, we extend the theory from Section 4.1, showing that the $v_m$, $\gamma_l$ and $\gamma_r$ computed as outlined there, are, in fact, jointly MLE-optimal.

Recall that an individual stump operating on feature $j_m$ is characterized by three parameters: $v_m$, $\gamma_l$ and $\gamma_r$. In Section 4.1, we show how to choose MLE-optimal $\gamma_l$ and $\gamma_r$ given $v_m$. It remains to show that if $v_m$ minimizes the entropy impurity $H_{\text{entropy}}$, then $\gamma_l^{\phi,\text{MLE}}$, $\gamma_l^{\phi,\text{MLE}}$, and $v_m$ are jointly MLE-optimal.

For an arbitrary split position $v_m$, we have the probabilities $p_{l,i}(v_m) = \mathbb{P}_{\boldsymbol{x}' \sim \phi(\boldsymbol{x}_i)}[x'_{j_m} \leq v_m]$ and $p_{r,i}(v_m) = 1 - p_{l,i}(v_m)$ of $\boldsymbol{x}'_i$ lying to the left or the right of $v_m$, respectively, under the input randomization scheme $\phi$. For an i.i.d. dataset with $n$ samples $(\boldsymbol{x}_i, y_i) \sim (\mathcal{X}, \mathcal{Y})$, we define the probabilities $p_j^y(v_m) = \frac{1}{n} \sum_{\{i | y_i = y\}} p_{j,i}(v_m)$ of picking the $j \in \{l, r\}$ leaf, conditioned on the target label, and $p_j(v_m) = p_j^0(v_m) + p_j^1(v_m)$ as their sum. Now, we compute the entropy impurity $H_{\text{entropy}}$ [32] as

$$H_{\text{entropy}}(v_m) = -\sum_{j \in \{l,r\}} p_j(v_m) \sum_{y \in \{0,1\}} \frac{p_j^y(v_m)}{p_j(v_m)} \log\left(\frac{p_j^y(v_m)}{p_j(v_m)}\right)$$

$$= -\sum_{j \in \{l,r\}} \sum_{y \in \{0,1\}} p_j^y(v_m) \log\left(\frac{p_j^y(v_m)}{p_j(v_m)}\right).$$

Similarly, let $\gamma_l^{\phi,\text{MLE}}(v_m)$ and $\gamma_l^{\phi,\text{MLE}}(v_m)$ be the MLE-optimal predictions given $v_m$, as computed in Section 4.1. We formalize our statement as follows in Theorem A.1:

**Theorem A.1.** *Given an i.i.d. dataset with $n$ samples $(\boldsymbol{x}_i, y_i) \sim (\mathcal{X}, \mathcal{Y})$, let $v_m^* := \arg\min_{v_m} H_{\text{entropy}}(v_m)$, $\gamma_l(v_m^*) = \frac{p_l^1(v_m^*)}{p_l^1(v_m^*) + p_l^0(v_m^*)}$ and $\gamma_r(v_m^*) = \frac{p_r^1(v_m^*)}{p_r^1(v_m^*) + p_r^0(v_m^*)}$. Then $v_m^*, \gamma_l$ and $\gamma_r$ are jointly MLE-optimal with respect to that dataset.*

*Proof.* Similarly to Section 4.1, but also optimizing over $v_m$, we obtain:

$$v_m^{\phi\text{MLE}}, \gamma_l^{\phi\text{MLE}}, \gamma_r^{\phi\text{MLE}} = \underset{v_m, \gamma_l, \gamma_r}{\arg\max} \; \mathbb{P}[\mathcal{Y} \mid \phi(\mathcal{X}), f_m]$$

$$= \underset{v_m, \gamma_l, \gamma_r}{\arg\max} \sum_{i=1}^{n} \mathbb{E}_{\boldsymbol{x}' \sim \phi(\boldsymbol{x}_i)} \left[\log \mathbb{P}[y_i \mid \boldsymbol{x}', f_m]\right]$$

$$= \underset{v_m, \gamma_l, \gamma_r}{\arg\max} \sum_{i \in \{i | y_i = 0\}}^{n} p_{l,i}(v_m) \log(1 - \gamma_l) + p_{r,i}(v_m) \log(1 - \gamma_r)$$

$$+ \sum_{i \in \{i | y_i = 1\}}^{n} p_{l,i}(v_m) \log(\gamma_l) + p_{r,i}(v_m) \log(\gamma_r)$$

$$= \underset{v_m, \gamma_l, \gamma_r}{\arg\max} \ p_l^0(v_m) \log(1 - \gamma_l) + p_r^0(v_m) \log(1 - \gamma_r)$$
$$+ \ p_l^1(v_m) \log(\gamma_l) + p_r^1(v_m) \log(\gamma_r)$$

As shown in Section 4.1, for a fixed $v_m$, the MLE-optimal estimates for $\gamma_l$ and $\gamma_r$ are $\gamma_l^{\phi\text{MLE}}(v_m) = \frac{p_l^1(v_m)}{p_l^1(v_m) + p_l^0(v_m)}$ and $\gamma_r^{\phi\text{MLE}}(v_m) = \frac{p_r^1(v_m)}{p_r^1(v_m) + p_r^0(v_m)}$. Hence, in the following, it is enough to optimize over $v_m$, substituting in $\gamma_l^{\phi\text{MLE}}(v_m)$ and $\gamma_r^{\phi\text{MLE}}(v_m)$. We obtain:

$$v_m^{\phi\text{MLE}} = \underset{v_m}{\arg\max} \ p_l^0(v_m) \log(1 - \gamma_l^{\phi,\text{MLE}}(v_m)) + p_r^0(v_m) \log(1 - \gamma_r^{\phi,\text{MLE}}(v_m))$$
$$+ \ p_l^1(v_m) \log(\gamma_l^{\phi,\text{MLE}}(v_m)) + p_r^1(v_m) \log(\gamma_r^{\phi,\text{MLE}}(v_m))$$
$$= \underset{v_m}{\arg\max} \ p_l^0(v_m) \log\left(1 - \frac{p_l^1(v_m)}{p_l^1(v_m) + p_l^0(v_m)}\right) + p_r^0(v_m) \log\left(1 - \frac{p_r^1(v_m)}{p_r^1(v_m) + p_r^0(v_m)}\right)$$
$$+ \ p_l^1(v_m) \log\left(\frac{p_l^1(v_m)}{p_l^1(v_m) + p_l^0(v_m)}\right) + p_r^1(v_m) \log\left(\frac{p_r^1(v_m)}{p_r^1(v_m) + p_r^0(v_m)}\right)$$
$$= \underset{v_m}{\arg\max} \ p_l^0(v_m) \log\left(1 - \frac{p_l^1(v_m)}{p_l(v_m)}\right) + p_r^0(v_m) \log\left(1 - \frac{p_r^1(v_m)}{p_r(v_m)}\right)$$
$$+ \ p_l^1(v_m) \log\left(\frac{p_l^1(v_m)}{p_l(v_m)}\right) + p_r^1(v_m) \log\left(\frac{p_r^1(v_m)}{p_r(v_m)}\right)$$
$$= \underset{v_m}{\arg\max} \ -H_{\text{entropy}}(v_m)$$
$$= \underset{v_m}{\arg\min} \ H_{\text{entropy}}(v_m)$$
$$= v_m^*$$

Thus, we have that the triplet $v_m^* := \arg\min_{v_m} H_{\text{entropy}}(v_m)$, $\gamma_l(v_m^*) = \frac{p_l^1(v_m^*)}{p_l^1(v_m^*) + p_l^0(v_m^*)}$ and $\gamma_r(v_m^*) = \frac{p_r^1(v_m^*)}{p_r^1(v_m^*) + p_r^0(v_m^*)}$ is jointly MLE-optimal. $\qquad\square$

### A.3 Gradient Boosting for Certifiable Robustness

Below, we describe ROBTREEBOOST, already outlined in Section 4.2, in more detail. Formally, we aim to minimize the cross-entropy loss between the certifiable prediction at the $q^{\text{th}}$ percentile $\mathcal{F}_{m-1,\boldsymbol{x}_i}^{-1}(q)$ and the one-hot target probability given by the label $y$, where we choose $q = \rho^{-1}(r)$ for some target radius r. Concretely, to add the $m^{\text{th}}$ stump to our ensemble, we begin by computing the certifiable prediction $y_i'$:

$$y_i' = \begin{cases} \bar{\mathcal{F}}_{m-1,\boldsymbol{x}_i}^{-1}(q) & \text{if } y = 0 \\ \mathcal{F}_{m-1,\boldsymbol{x}_i}^{-1}(1-q) & \text{if } y = 1. \end{cases} \tag{5}$$

Now, we define the pseudo label $\tilde{y}$ as the residual between the target label $y$ and the certifiable prediction $y'$, scaled to $[0, 1]$ as $\tilde{y}_i = \frac{1}{2} + \frac{y_i - y_i'}{2}$. Subsequently, we select feature $j_m$ and split position $v_m$ that minimize the mean squared error impurity (MSE) under the randomization scheme for these pseudo-labels. As before, we define the mean squared error impurity $H_{\text{MSE}}$ in terms of the branching probabilities $p_{l,i} = \mathbb{P}_{\boldsymbol{x}' \sim \phi(\boldsymbol{x}_i)}[x_{j_m}' \leq v_m]$ and $p_{r,i} = 1 - p_{l,i}$:

$$\mu_j = \frac{\sum_{i=1}^n p_{j,i} \tilde{y}_i}{\sum_{i=1}^n p_{j,i}} \qquad\qquad H_{\text{MSE}} = \frac{\sum_{i=1}^n \sum_{j \in \{l,r\}} p_{j,i} (\tilde{y}_i - \mu_j)^2}{n}. \tag{6}$$

The optimal leaf predictions can now be computed approximately [33] to

$$\gamma_l = \frac{\sum_{i=1}^n p_{l,i} \tilde{y}_i'}{\sum_i p_{l,i} |2\tilde{y}_i' - 1|(1 - |2\tilde{y}_i' - 1|)}, \tag{7}$$

and $\gamma_r$ analogously. We initialize this boosting process with an ensemble of individually MLE-optimal stumps and repeat this boosting step until we have added as many stumps as desired.

## A.4 Adaptive Boosting for Certifiable Robustness

Below, we describe ROBADABOOST, already outlined in Section 4.2, in more detail. Our goal is to obtain a weighted ensemble $\bar{F}_K$

$$\bar{F}_K(\boldsymbol{x}) = \frac{1}{\sum_{k=1}^{K} \alpha^k} \sum_{k=1}^{K} \alpha^k \mathbb{1}_{\mathbb{P}_{\boldsymbol{x}' \sim \phi(\boldsymbol{x})}[\bar{f}_M^k(\boldsymbol{x}') > 0.5] > 0.5}, \tag{8}$$

consisting of $K$ stump ensembles $\bar{f}_M^k$, that is certifiably robust at a pre-determined radius $r$. Here, $\bar{F}_K(\boldsymbol{x})\colon \mathbb{R}^d \to [0, 1]$ is a soft-classifier, that predicts class 1 for outputs $> 0.5$ and class 0 else.

To train the $K$ constituting ensembles such that the overall ensemble $\bar{F}_K$ is certifiably robust at radius $r$, we proceed as follows: First, we initialize the weights of all samples $\boldsymbol{x}_i$ to $w_i^1 = \frac{1}{n}$. Then, for $k = 1$ to $K$, we iteratively fit a new stump ensemble $\bar{f}_M^k$ as described in Section 4.1 using the sample weights $w_i^k$. Then, similar to Freund and Schapire [34] although targeting certifiability instead of accuracy, we update the sample weights as follows: First, we compute whether the newly trained $k$-th ensemble $\bar{f}_M^k$ is certifiably correct ($c_i$) for each sample $\boldsymbol{x}_i$ in the training set:

$$c_i = \begin{cases} \mathbb{1}_{\mathbb{P}_{\boldsymbol{x}' \sim \phi(\boldsymbol{x}_i)}[\bar{f}_M^k(\boldsymbol{x}') \leq 0.5] > \rho_{\boldsymbol{x}}^{-1}(r)} & \text{if } y = 0 \\ \mathbb{1}_{\mathbb{P}_{\boldsymbol{x}' \sim \phi(\boldsymbol{x}_i)}[\bar{f}_M^k(\boldsymbol{x}') > 0.5] > \rho_{\boldsymbol{x}}^{-1}(r)} & \text{if } y = 1. \end{cases} \tag{9}$$

Then, we determine the certifiable error $err^k$, and the model weight $\alpha^k$ of $f_m^k$ as:

$$err^k = \frac{\sum_{i=1}^{n} w_i(1 - c_i)}{\sum_{i=1}^{n} w_i} \qquad \alpha^k = \log \frac{1 - err^k}{err^k}$$

and update the sample weights for the next iteration to:

$$w_i^{k+1} = \frac{w_i^k \exp(\alpha^k(1 - c_i))}{\sum_{i=1}^{n} w_i^k \exp(\alpha^k(1 - c_i))}$$

before training the next ensemble. This way, we are minimizing the overall loss for certified predictions at radius $r$.

To certify $\bar{F}_K$ at a specific radius $r$, we now have to show that we can certify individual ensembles corresponding to at least half the total weights, or more formally (here, without loss of generality assuming a label of $y = 1$):

$$\sum_{k=1}^{K} \alpha^k \mathbb{1}_{\mathbb{P}_{\boldsymbol{x}' \sim \phi(\boldsymbol{x})}[\bar{f}_M^k(\boldsymbol{x}') > 0.5] > \rho_{\boldsymbol{x}}^{-1}(r)} > \frac{\sum_{k=1}^{K} |\alpha^k|}{2}. \tag{10}$$

To compute the certifiable radius for $\bar{F}_k$, we compute the certifiable radii $R^k$ of the individual ensembles, sort them in decreasing order such that $R^k \geq R^{k+1}$ and obtain the largest radius $R^k$ such that $\sum_{l=1}^{k} \alpha^l > \frac{\sum_{l=1}^{K} |\alpha^l|}{2}$. Intuitively, we need to find a subset of models such that their weighted predictions for class 1 reach at least half the possible weight, accounting for negative weights.

# B  Experimental Details

Here, we describe our experimental setup in greater detail. Note that we also publish all code, models, and instructions required to reproduce our results at `https://github.com/eth-sri/drs`.

## B.1  Datasets

In this section, we describe the datasets we use in detail.

**Datasets with Numerical Features** We conduct experiments focusing on numerical features only on all the datasets considered by prior work [23, 22]. More concretely, we use the tabular datasets BREASTCANCER [37] and DIABETES [36], where we follow prior work [23] in using the first 80% of the samples as train set and the remaining 20% as test set, normalizing the data to $[0, 1]$, and the

vision datasets MNIST 1 vs. 5 [39], MNIST 2 vs. 6 [39], and FMNIST-SHOES [38], where we use all samples of the right classes from the train and test sets.

Additionally, we consider the SPAMBASE [37] dataset, where the task is to predict whether an email is spam (binary classification) given 57 numerical features. We normalize all features using the mean and standard deviation of the training data before applying any perturbations.

**Datasets with Numerical and Categorical Features** We conduct our experiments on the joint certification of numerical and categorical features using the popular ADULT [37], CREDIT [37], MAMMO [37], and BANK [37] datasets. By default, we use the first 70% of the samples as the train set, and the remaining 30% as the test set. For error bound experiments (in App. C.1), we use 5-fold cross-validation over the whole datasets, and report the mean and standard deviations over the 5 folds. Here, we normalize the numerical features using the mean and standard deviation of the training data, before applying any perturbations.

The ADULT [37] dataset is a societal dataset based on the 1994 US Census database. It contains eight categorical and six numerical variables for each individual. The cardinalities of the categorical variables range from 2 to 42 (concretely, they are $9, 16, 7, 15, 6, 5, 2$, and $42$). The task is to predict whether an individual's salary is below or above 50k USD.

The CREDIT [37] dataset is a financial dataset containing 13 categorical and 7 numerical features. The cardinalities of the categorical features range from 2 to 10 (concretely, they are $4, 5, 10, 5, 5, 4, 3, 4, 3, 3, 4, 2$, and $2$). The task is to predict whether a customer has a low or high risk to default on a loan.

The MAMMO [37] dataset is medial dataset where the goal is to predicting whether breast biopsies are needed. It consists of 3 numerical and 2 categorical where the categorical features have cardinalities 4 and 5.

The BANK [37] dataset is a financial dataset, consisting of 9 categorical and 7 numerical features. The task is to predict whether a client will subscribe to a bank term deposit or not, given the features.

Some datasets exhibit a significant class imbalance, with the minority class constituting 24.6% of the ADULT and 29.6% of the CREDIT train set. Therefore, we report balanced certified accuracy, computed as the arithmetic mean of the per class certified accuracies.

**Dataset with Categorical Features** The MUSHROOM [37] dataset contains 22 categorical features encoding physical features of mushrooms with the goal to predicting whether a mushroom is edible or poisonous.

## B.2 Training Details

The key (and for independent training, the only) hyper-parameter of our approach is the noise magnitude, $\lambda$ for $\ell_1$-certification and $\sigma$ for $\ell_2$-certification. In Table 5, we report the noise levels chosen for the different datasets. We discuss the effect of different noise magnitudes in App. C.4 and observe that results are generally quite stable across a wide range of noise magnitudes. Unless otherwise stated, we determine the split position $v_m$ via linear search using increments of size 0.01 and discretize leaf predictions $\gamma$ using 100 steps (i.e., $\Delta = 100$).

**ROBTREEBOOST** We initialize ROBTREE-BOOST with an ensemble of independently trained stumps and add a further $n_b$ stumps as described in Section 4.2 using the $q^{\text{th}}$ percentile to compute the certifiable predictions. We chose $q$ and $n_b$ as shown in Table 6.

Table 5: Noise magnitudes used for Table 2.

| Method | Dataset | $\lambda$ (for $\ell_1$) | $\sigma$ (for $\ell_2$) |
|---|---|---|---|
| | BREASTCANCER | 2.00 | 4.00 |
| | DIABETES | 0.35 | 0.25 |
| Independent | MNIST 1 vs. 5 | 4.00 | 0.25 |
| | MNIST 2 vs. 6 | 4.00 | 0.25 |
| | FMNIST-SHOES | 4.00 | 0.25 |
| | BREASTCANCER | 2.00 | 0.25 |
| | DIABETES | 0.28 | 0.15 |
| Boosting | MNIST 1 vs. 5 | 4.00 | 0.25 |
| | MNIST 2 vs. 6 | 4.00 | 0.25 |
| | FMNIST-SHOES | 4.00 | 0.25 |

Table 6: ROBTREEBOOST parameters.

| Parameter | Perturbation | BREASTCANCER | DIABETES |
|---|---|---|---|
| Percentile $q$ | $\ell_1$ | 0.60 | 0.70 |
| | $\ell_2$ | 0.98 | 0.95 |
| Additional stumps $n_b$ | $\ell_1$ | 30 | 15 |
| | $\ell_2$ | 40 | 100 |

**ROBADABOOST** To evaluate ROBADABOOST, we consider ensembles of $K = 20$ individual stump ensembles in each of our experiments. We choose the same noise magnitudes as for independently trained stumps, described in Table 5.

**Joint Certification** For joint certification, we use ensembles of independently trained decision stumps, one for each feature. The stump corresponding to categorical features maps a categorical value to either $0.375$ or $0.625$ (which are the same distance from the decision threshold $0.5$), depending on whether the majority of the samples with this categorical value have class 0 or 1, respectively. Note that permitting arbitrary leaf predictions slightly improves clean accuracy, but significantly worsens worst-case behaviour. Choosing leaf predictions further from the decision threshold gives more emphasis to categorical variables compared to numerical ones. The stumps for the numerical features are learned individually, as described in Section 4.1. For $\ell_1$, we used the noise magnitude $\lambda = 2.0$ and for $\ell_2$-certification $\sigma = 0.25$.

### B.3 Computational Resources and Experimental Timings

In this section, we describe the computational resources required for our experiments. We run all our experiments using 24 cores of an Intel Xeon Gold 6242 CPUs and a single NVIDIA RTX 2080Ti and report timings for the full experiment in App. B.3. We show timings in Table 7.

We observe that all certification is extremely quick with FMNIST-SHOES taking the longest at 6s for the whole test set and an ensemble of independently trained stumps in the $\ell_1$-setting, translating to 0.003s per sample. When evaluating models in single instead of double precision, we can, e.g., further reduce certification times from 3s to 1.2s for MNIST 2 VS. 6. The independently MLE-optimal training is similarly quick, allowing us to run all core experiments in

Table 7: Experimental timings for whole datasets.

| Norm | Dataset | Independent | | Boosting | |
|---|---|---|---|---|---|
| | | Training | Certification | Training | Certification |
| $\ell_1$ | BREASTCANCER | 2s | $< 0.1$s | 14s | $< 0.1$s |
| | DIABETES | 2s | $< 0.1$s | 2s | $< 0.1$s |
| | MNIST 1 VS. 5 | 32s | 5s | 13min | 27s |
| | MNIST 2 VS. 6 | 29s | 4s | 11min | 16s |
| | FMNIST-SHOES | 31s | 6s | 13min | 40s |
| $\ell_2$ | BREASTCANCER | 2s | $< 0.1$s | 9s | $< 0.1$s |
| | DIABETES | 2s | $< 0.1$s | 47s | $< 0.1$s |
| | MNIST 1 VS. 5 | 14s | 4s | 10min | 29s |
| | MNIST 2 VS. 6 | 14s | 3s | 9min | 26s |
| | FMNIST-SHOES | 15s | 4s | 9min | 27s |

less than 5 minutes. Only ROBADABOOST takes more than one minute for an individual experiment, as it involves training and certifying 20 stump ensembles. For datasets combining categorical and numerical features, the training and certification for the categorical variables is almost instantaneous and dominated by that for the numerical features. The latter requires 19.9s and 47.0s for the $\ell_1$ and $\ell_2$-experiment, respectively, on ADULT and 1.5s respectively 2.0s on CREDIT. We remark that computational efficiency was not a main focus of this work and we did not optimize runtimes.

## C   Additional Experiments

In this section, we extend our experimental evaluation from Section 5. Concretely, in App. C.1, we provide additional experiments on the joint certification of categorical and numerical variables. In App. C.2, we compare DRS to RS in more detail while in App. C.3, we continue our investigation of our MLE optimality criterion. Moreover, in App. C.4, we provide additional experiments on the effect of the noise magnitudes $\lambda$ and $\sigma$ for $\ell_1$- and $\ell_2$-certification, respectively. In App. C.5, we analyze the impact of the discretization granularity and in App. C.6, we evaluate the effect of an approximate split position optimization. Finally, in App. C.7, we include error bound experiments for certification of numerical features via 5-fold cross-validation.

### C.1   Additional Experiments on Joint Robustness Certificates

In Table 8, we report the mean and standard deviation (over a 5-fold cross-validation) of the balanced certified accuracies at a range of $\ell_2$ radii over the numerical features given varying perturbation levels of the categorical features for all datasets containing both numerical and categorical features. We report the corresponding imbalanced certified accuracies in Table 9 and similar results for $\ell_1$ radii in Tables 10 and 11.

We again observe that models utilizing both categorical and numerical features outperform those using only either one on clean data. Interestingly, the slower drop in certified accuracy with increasing perturbation of the numerical features is much more pronounced in the $\ell_1$-setting, and much higher certified accuracies are obtained even at large radii. For example, on ADULT, considering only

Table 8: Balanced certified accuracy (BCA) [%] under joint $\ell_0$- and $\ell_2$-perturbations of categorical and numerical features, respectively, depending on whether model uses categorical and/or numerical features. The balanced natural accuracy is the BCA at radius $r = 0.0$. Larger is better.

| Dataset | Categorical Features | $\ell_0$ Radius $r_0$ | BCA without Numerical Features | BCA with Numerical Features at $\ell_2$ Radius $r_2$ | | | | | | |
| --- | --- | --- | --- | --- | --- | --- | --- | --- | --- | --- |
| | | | | 0.00 | 0.25 | 0.50 | 0.75 | 1.00 | 1.25 | 1.50 |
| ADULT | no | - | - | $74.3_{\pm 0.4}$ | $65.5_{\pm 0.3}$ | $42.3_{\pm 0.5}$ | $26.9_{\pm 0.4}$ | $13.7_{\pm 0.3}$ | $7.7_{\pm 0.5}$ | $4.4_{\pm 0.3}$ |
| | yes | 0 | $76.2_{\pm 0.6}$ | $77.9_{\pm 0.4}$ | $74.2_{\pm 0.7}$ | $68.0_{\pm 0.6}$ | $62.9_{\pm 0.6}$ | $48.4_{\pm 0.4}$ | $39.6_{\pm 0.7}$ | $34.2_{\pm 0.4}$ |
| | | 1 | $57.0_{\pm 0.8}$ | $66.2_{\pm 0.8}$ | $61.5_{\pm 0.9}$ | $52.8_{\pm 0.7}$ | $45.9_{\pm 0.7}$ | $33.1_{\pm 0.4}$ | $25.0_{\pm 0.5}$ | $20.5_{\pm 0.4}$ |
| | | 2 | $32.9_{\pm 0.6}$ | $50.7_{\pm 0.6}$ | $45.4_{\pm 0.8}$ | $36.2_{\pm 0.5}$ | $27.8_{\pm 0.3}$ | $19.7_{\pm 0.3}$ | $15.2_{\pm 0.4}$ | $11.7_{\pm 0.3}$ |
| | | 3 | $8.9_{\pm 0.2}$ | $35.9_{\pm 0.5}$ | $30.8_{\pm 0.6}$ | $23.4_{\pm 0.4}$ | $14.6_{\pm 0.4}$ | $9.7_{\pm 0.4}$ | $7.2_{\pm 0.3}$ | $5.1_{\pm 0.2}$ |
| CREDIT | no | - | - | $59.4_{\pm 3.7}$ | $51.3_{\pm 3.0}$ | $39.6_{\pm 2.9}$ | $27.0_{\pm 5.3}$ | $19.5_{\pm 7.9}$ | $13.5_{\pm 6.3}$ | $6.7_{\pm 4.1}$ |
| | yes | 0 | $64.7_{\pm 4.2}$ | $65.3_{\pm 4.3}$ | $64.0_{\pm 4.9}$ | $62.1_{\pm 4.6}$ | $58.4_{\pm 4.0}$ | $53.3_{\pm 3.8}$ | $51.5_{\pm 4.6}$ | $49.2_{\pm 4.7}$ |
| | | 1 | $44.7_{\pm 3.5}$ | $48.2_{\pm 3.1}$ | $46.1_{\pm 3.1}$ | $42.1_{\pm 3.5}$ | $38.6_{\pm 3.5}$ | $35.4_{\pm 4.5}$ | $33.2_{\pm 5.2}$ | $31.1_{\pm 5.7}$ |
| | | 2 | $26.7_{\pm 5.7}$ | $29.8_{\pm 4.4}$ | $27.9_{\pm 4.4}$ | $24.5_{\pm 5.7}$ | $21.3_{\pm 5.8}$ | $18.7_{\pm 5.8}$ | $16.7_{\pm 5.6}$ | $15.5_{\pm 6.0}$ |
| | | 3 | $11.1_{\pm 4.3}$ | $14.2_{\pm 4.5}$ | $13.3_{\pm 3.8}$ | $11.4_{\pm 4.5}$ | $9.8_{\pm 3.6}$ | $8.5_{\pm 3.9}$ | $6.7_{\pm 3.7}$ | $6.5_{\pm 3.8}$ |
| MAMMO | no | - | - | $61.7_{\pm 2.8}$ | $61.6_{\pm 2.9}$ | $51.5_{\pm 3.0}$ | $13.8_{\pm 4.2}$ | $9.4_{\pm 5.0}$ | $8.6_{\pm 5.6}$ | $6.4_{\pm 6.3}$ |
| | yes | 0 | $79.0_{\pm 1.2}$ | $78.1_{\pm 2.6}$ | $78.1_{\pm 2.6}$ | $76.4_{\pm 2.0}$ | $51.1_{\pm 5.8}$ | $48.4_{\pm 9.5}$ | $47.6_{\pm 9.1}$ | $42.5_{\pm 12.0}$ |
| | | 1 | $30.8_{\pm 3.3}$ | $46.7_{\pm 6.2}$ | $46.6_{\pm 6.2}$ | $36.5_{\pm 6.9}$ | $11.3_{\pm 3.9}$ | $7.7_{\pm 4.0}$ | $7.1_{\pm 4.5}$ | $4.9_{\pm 5.1}$ |
| | | 2 | $0.0_{\pm 0.0}$ | $12.8_{\pm 2.2}$ | $12.7_{\pm 2.4}$ | $2.7_{\pm 2.0}$ | $2.6_{\pm 2.0}$ | $2.4_{\pm 2.0}$ | $2.1_{\pm 2.3}$ | $0.0_{\pm 0.0}$ |
| BANK | no | - | - | $73.1_{\pm 2.2}$ | $63.0_{\pm 1.4}$ | $47.7_{\pm 2.1}$ | $31.9_{\pm 1.4}$ | $17.9_{\pm 3.8}$ | $12.5_{\pm 5.0}$ | $7.4_{\pm 4.7}$ |
| | yes | 0 | $62.8_{\pm 1.9}$ | $69.9_{\pm 1.8}$ | $65.4_{\pm 1.4}$ | $57.0_{\pm 0.6}$ | $48.8_{\pm 0.7}$ | $39.6_{\pm 1.6}$ | $30.4_{\pm 1.6}$ | $24.0_{\pm 1.7}$ |
| | | 1 | $42.3_{\pm 1.5}$ | $53.6_{\pm 1.9}$ | $47.7_{\pm 2.1}$ | $40.1_{\pm 2.2}$ | $30.5_{\pm 2.0}$ | $21.7_{\pm 1.7}$ | $14.8_{\pm 1.9}$ | $9.8_{\pm 2.4}$ |
| | | 2 | $21.2_{\pm 2.3}$ | $37.4_{\pm 2.5}$ | $31.5_{\pm 2.1}$ | $23.2_{\pm 2.0}$ | $14.9_{\pm 2.0}$ | $9.0_{\pm 2.3}$ | $6.1_{\pm 2.2}$ | $4.3_{\pm 2.5}$ |
| | | 3 | $7.2_{\pm 2.3}$ | $21.8_{\pm 2.9}$ | $17.5_{\pm 2.7}$ | $11.0_{\pm 2.3}$ | $5.6_{\pm 1.3}$ | $3.0_{\pm 1.4}$ | $2.2_{\pm 1.4}$ | $1.0_{\pm 0.4}$ |

Table 9: Certified accuracy (CA) [%] under joint $\ell_0$- and $\ell_2$-perturbations of categorical and numerical features, respectively, depending on whether model uses categorical and/or numerical features. The natural accuracy is the CA at radius $r = 0.0$. Larger is better.

| Dataset | Categorical Features | $\ell_0$ Radius $r_0$ | CA without Numerical Features | CA with Numerical Features at $\ell_2$ Radius $r_2$ | | | | | | |
| --- | --- | --- | --- | --- | --- | --- | --- | --- | --- | --- |
| | | | | 0.00 | 0.25 | 0.50 | 0.75 | 1.00 | 1.25 | 1.50 |
| ADULT | no | - | - | $74.2_{\pm 0.5}$ | $65.7_{\pm 0.6}$ | $37.5_{\pm 1.0}$ | $23.2_{\pm 0.8}$ | $9.6_{\pm 0.3}$ | $4.0_{\pm 0.5}$ | $2.1_{\pm 0.2}$ |
| | yes | 0 | $69.7_{\pm 0.6}$ | $70.0_{\pm 0.5}$ | $65.8_{\pm 0.7}$ | $58.6_{\pm 0.7}$ | $53.7_{\pm 0.7}$ | $34.9_{\pm 0.7}$ | $24.4_{\pm 0.8}$ | $19.1_{\pm 0.4}$ |
| | | 1 | $52.0_{\pm 0.8}$ | $58.3_{\pm 0.9}$ | $53.6_{\pm 1.0}$ | $43.8_{\pm 0.8}$ | $36.9_{\pm 0.8}$ | $21.3_{\pm 0.5}$ | $13.0_{\pm 0.6}$ | $10.2_{\pm 0.3}$ |
| | | 2 | $27.5_{\pm 0.6}$ | $43.1_{\pm 0.7}$ | $38.3_{\pm 0.8}$ | $28.2_{\pm 0.6}$ | $19.4_{\pm 0.4}$ | $11.0_{\pm 0.4}$ | $7.5_{\pm 0.3}$ | $5.7_{\pm 0.2}$ |
| | | 3 | $6.6_{\pm 0.2}$ | $28.7_{\pm 0.5}$ | $24.1_{\pm 0.7}$ | $16.5_{\pm 0.3}$ | $8.4_{\pm 0.3}$ | $4.8_{\pm 0.3}$ | $3.5_{\pm 0.2}$ | $2.4_{\pm 0.1}$ |
| CREDIT | no | - | - | $63.7_{\pm 3.5}$ | $55.3_{\pm 3.9}$ | $43.5_{\pm 4.6}$ | $30.7_{\pm 6.0}$ | $21.7_{\pm 9.1}$ | $14.0_{\pm 7.2}$ | $7.7_{\pm 5.2}$ |
| | yes | 0 | $58.3_{\pm 9.5}$ | $59.3_{\pm 9.5}$ | $57.9_{\pm 10.1}$ | $55.8_{\pm 10.2}$ | $52.4_{\pm 9.6}$ | $48.2_{\pm 8.7}$ | $45.8_{\pm 9.2}$ | $43.1_{\pm 9.4}$ |
| | | 1 | $38.2_{\pm 7.4}$ | $42.4_{\pm 8.5}$ | $40.4_{\pm 8.5}$ | $36.5_{\pm 9.0}$ | $33.5_{\pm 8.2}$ | $30.8_{\pm 8.0}$ | $28.4_{\pm 7.8}$ | $26.1_{\pm 7.2}$ |
| | | 2 | $21.7_{\pm 5.0}$ | $25.0_{\pm 6.0}$ | $23.0_{\pm 5.9}$ | $20.4_{\pm 5.5}$ | $17.7_{\pm 5.4}$ | $15.3_{\pm 4.7}$ | $13.6_{\pm 5.0}$ | $12.3_{\pm 4.7}$ |
| | | 3 | $8.4_{\pm 2.6}$ | $11.0_{\pm 2.5}$ | $10.3_{\pm 2.3}$ | $8.8_{\pm 2.8}$ | $7.5_{\pm 2.4}$ | $6.4_{\pm 2.2}$ | $5.1_{\pm 2.5}$ | $4.9_{\pm 2.5}$ |
| MAMMO | no | - | - | $61.7_{\pm 3.7}$ | $61.6_{\pm 3.9}$ | $51.6_{\pm 3.4}$ | $13.8_{\pm 4.4}$ | $9.6_{\pm 5.3}$ | $8.8_{\pm 6.0}$ | $6.6_{\pm 6.7}$ |
| | yes | 0 | $79.0_{\pm 1.3}$ | $77.9_{\pm 2.8}$ | $77.9_{\pm 2.8}$ | $76.3_{\pm 2.2}$ | $51.2_{\pm 5.0}$ | $48.7_{\pm 8.1}$ | $47.8_{\pm 7.8}$ | $42.6_{\pm 11.4}$ |
| | | 1 | $31.8_{\pm 4.5}$ | $46.6_{\pm 6.2}$ | $46.5_{\pm 6.2}$ | $36.5_{\pm 6.9}$ | $11.3_{\pm 4.0}$ | $7.8_{\pm 4.3}$ | $7.2_{\pm 4.8}$ | $5.1_{\pm 5.5}$ |
| | | 2 | $0.0_{\pm 0.0}$ | $12.8_{\pm 2.0}$ | $12.6_{\pm 2.1}$ | $2.6_{\pm 1.9}$ | $2.5_{\pm 1.9}$ | $2.4_{\pm 1.9}$ | $2.0_{\pm 2.2}$ | $0.0_{\pm 0.0}$ |
| BANK | no | - | - | $68.6_{\pm 2.3}$ | $56.9_{\pm 2.1}$ | $41.1_{\pm 1.3}$ | $21.2_{\pm 2.0}$ | $5.6_{\pm 0.9}$ | $3.1_{\pm 1.3}$ | $1.7_{\pm 1.1}$ |
| | yes | 0 | $64.4_{\pm 7.5}$ | $73.8_{\pm 5.8}$ | $68.6_{\pm 6.3}$ | $58.9_{\pm 7.2}$ | $47.1_{\pm 8.0}$ | $33.8_{\pm 7.4}$ | $20.7_{\pm 6.0}$ | $13.4_{\pm 4.2}$ |
| | | 1 | $44.4_{\pm 7.9}$ | $58.8_{\pm 6.7}$ | $52.4_{\pm 6.7}$ | $41.7_{\pm 7.6}$ | $29.3_{\pm 7.3}$ | $16.9_{\pm 5.8}$ | $8.1_{\pm 2.6}$ | $3.8_{\pm 0.9}$ |
| | | 2 | $23.2_{\pm 7.1}$ | $41.9_{\pm 6.9}$ | $35.9_{\pm 6.7}$ | $24.8_{\pm 6.6}$ | $13.4_{\pm 4.9}$ | $6.0_{\pm 2.7}$ | $2.6_{\pm 0.8}$ | $1.1_{\pm 0.6}$ |
| | | 3 | $8.3_{\pm 4.7}$ | $24.9_{\pm 5.7}$ | $19.3_{\pm 5.2}$ | $10.7_{\pm 3.7}$ | $4.7_{\pm 1.9}$ | $1.6_{\pm 0.6}$ | $0.6_{\pm 0.3}$ | $0.2_{\pm 0.1}$ |

numerical features leads to a BCA of $62.1\%_{\pm 0.3\%}$ at radius $r_1 = 0.0$ dropping to $42.0\%_{\pm 0.3\%}$ at $r_1 = 1.5$. In contrast, when also utilizing categorical features, the BCA at $r_1 = 0.0$ is $76.9\%_{\pm 0.5\%}$, only dropping to $71.1\%_{\pm 0.6\%}$ at $r_1 = 1.5$, when no categorical variable is perturbed ($r_0 = 0$). Similarly, when at most one categorical variable is perturbed, the BCA at $r_1 = 0.0$ is $59.7\%_{\pm 0.6\%}$ and only drops to $54.4\%_{\pm 0.8\%}$ radius $r_1 = 1.5$. This highlights again that, when available, utilizing categorical features in addition to numerical ones is essential to improve accuracy and make models more certifiably robust.

While standard deviations are generally moderately low, the sensitivity to different train/test-splits is particularly small for datasets with many samples like ADULT (nearly $50'000$ samples).

DRS is also applicable to data sets involving only categorical features as ca be seen in Table 12, where we report results on MUSHROOM. As expected, we observe that both balanced and imbalanced certifiable accuracy decrease as we permit more and more categorical features to be perturbed.

Table 10: Balanced certified accuracy (BCA) [%] under joint $\ell_0$- and $\ell_1$-perturbations of categorical and numerical features, respectively, depending on whether model uses categorical and/or numerical features. The balanced natural accuracy is the BCA at radius $r = 0.0$. Larger is better.

| Dataset | Categorical Features | $\ell_0$ Radius $r_0$ | BCA without Numerical Features | BCA with Numerical Features at $\ell_1$ Radius $r_1$ | | | | | | |
|---|---|---|---|---|---|---|---|---|---|---|
| | | | | 0.00 | 0.25 | 0.50 | 0.75 | 1.00 | 1.25 | 1.50 |
| ADULT | no | - | - | $62.1_{\pm 0.3}$ | $58.8_{\pm 0.5}$ | $54.9_{\pm 0.4}$ | $51.5_{\pm 0.5}$ | $47.1_{\pm 0.5}$ | $44.6_{\pm 0.4}$ | $42.0_{\pm 0.3}$ |
| | yes | 0 | $76.2_{\pm 0.6}$ | $76.9_{\pm 0.5}$ | $76.4_{\pm 0.5}$ | $75.6_{\pm 0.5}$ | $74.8_{\pm 0.6}$ | $73.6_{\pm 0.6}$ | $72.6_{\pm 0.6}$ | $71.1_{\pm 0.6}$ |
| | | 1 | $57.0_{\pm 0.8}$ | $59.7_{\pm 0.6}$ | $59.1_{\pm 0.6}$ | $58.4_{\pm 0.6}$ | $57.6_{\pm 0.7}$ | $56.8_{\pm 0.7}$ | $55.8_{\pm 0.8}$ | $54.4_{\pm 0.8}$ |
| | | 2 | $32.9_{\pm 0.6}$ | $38.3_{\pm 0.5}$ | $37.3_{\pm 0.5}$ | $36.3_{\pm 0.4}$ | $35.6_{\pm 0.4}$ | $34.8_{\pm 0.5}$ | $34.2_{\pm 0.5}$ | $33.3_{\pm 0.5}$ |
| | | 3 | $8.9_{\pm 0.2}$ | $17.0_{\pm 0.3}$ | $15.3_{\pm 0.3}$ | $14.3_{\pm 0.2}$ | $13.3_{\pm 0.3}$ | $12.3_{\pm 0.3}$ | $11.9_{\pm 0.3}$ | $11.5_{\pm 0.2}$ |
| CREDIT | no | - | - | $58.2_{\pm 4.0}$ | $54.7_{\pm 4.0}$ | $51.4_{\pm 3.4}$ | $48.0_{\pm 2.6}$ | $43.0_{\pm 1.3}$ | $33.3_{\pm 1.5}$ | $26.9_{\pm 2.2}$ |
| | yes | 0 | $64.7_{\pm 4.2}$ | $65.1_{\pm 4.4}$ | $64.5_{\pm 4.1}$ | $64.0_{\pm 3.8}$ | $63.4_{\pm 3.5}$ | $62.8_{\pm 3.6}$ | $61.2_{\pm 4.6}$ | $60.6_{\pm 4.9}$ |
| | | 1 | $44.7_{\pm 3.5}$ | $46.1_{\pm 3.3}$ | $45.4_{\pm 2.8}$ | $45.2_{\pm 3.0}$ | $44.5_{\pm 3.3}$ | $43.9_{\pm 2.7}$ | $43.3_{\pm 2.9}$ | $42.2_{\pm 3.3}$ |
| | | 2 | $26.7_{\pm 5.7}$ | $28.1_{\pm 5.6}$ | $27.8_{\pm 5.5}$ | $27.1_{\pm 5.9}$ | $26.2_{\pm 6.0}$ | $26.1_{\pm 6.0}$ | $25.3_{\pm 6.7}$ | $24.4_{\pm 6.4}$ |
| | | 3 | $11.1_{\pm 4.3}$ | $12.7_{\pm 5.1}$ | $12.6_{\pm 4.9}$ | $11.9_{\pm 4.8}$ | $11.5_{\pm 4.8}$ | $11.0_{\pm 4.2}$ | $10.6_{\pm 4.3}$ | $10.2_{\pm 4.1}$ |
| MAMMO | no | - | - | $51.0_{\pm 1.2}$ | $49.3_{\pm 1.2}$ | $48.4_{\pm 1.3}$ | $48.4_{\pm 1.3}$ | $48.1_{\pm 1.1}$ | $45.9_{\pm 1.3}$ | $45.8_{\pm 1.2}$ |
| | yes | 0 | $79.0_{\pm 1.2}$ | $77.0_{\pm 1.9}$ | $76.8_{\pm 1.8}$ | $76.6_{\pm 1.9}$ | $76.6_{\pm 1.9}$ | $76.6_{\pm 1.9}$ | $74.4_{\pm 2.4}$ | $74.4_{\pm 2.4}$ |
| | | 1 | $30.8_{\pm 3.3}$ | $41.0_{\pm 3.8}$ | $39.5_{\pm 3.2}$ | $38.9_{\pm 3.2}$ | $38.9_{\pm 3.2}$ | $38.6_{\pm 3.3}$ | $37.2_{\pm 3.8}$ | $37.0_{\pm 4.0}$ |
| | | 2 | $0.0_{\pm 0.0}$ | $0.0_{\pm 0.0}$ | $0.0_{\pm 0.0}$ | $0.0_{\pm 0.0}$ | $0.0_{\pm 0.0}$ | $0.0_{\pm 0.0}$ | $0.0_{\pm 0.0}$ | $0.0_{\pm 0.0}$ |
| BANK | no | - | - | $69.5_{\pm 2.5}$ | $64.7_{\pm 2.0}$ | $61.1_{\pm 1.8}$ | $56.7_{\pm 1.7}$ | $52.1_{\pm 1.7}$ | $47.3_{\pm 1.9}$ | $41.3_{\pm 1.3}$ |
| | yes | 0 | $62.8_{\pm 1.9}$ | $68.3_{\pm 0.6}$ | $66.4_{\pm 0.7}$ | $64.9_{\pm 0.7}$ | $63.2_{\pm 0.9}$ | $61.2_{\pm 0.5}$ | $59.0_{\pm 0.9}$ | $55.9_{\pm 1.7}$ |
| | | 1 | $42.3_{\pm 1.5}$ | $49.1_{\pm 1.6}$ | $46.9_{\pm 1.4}$ | $44.8_{\pm 1.3}$ | $43.3_{\pm 1.6}$ | $40.6_{\pm 1.6}$ | $38.9_{\pm 1.9}$ | $36.3_{\pm 1.9}$ |
| | | 2 | $21.2_{\pm 2.3}$ | $30.9_{\pm 1.6}$ | $29.0_{\pm 1.6}$ | $27.9_{\pm 2.2}$ | $25.9_{\pm 2.4}$ | $23.9_{\pm 1.9}$ | $22.4_{\pm 1.8}$ | $20.2_{\pm 1.9}$ |
| | | 3 | $7.2_{\pm 2.3}$ | $15.5_{\pm 2.6}$ | $14.3_{\pm 2.9}$ | $13.0_{\pm 2.8}$ | $12.0_{\pm 2.8}$ | $10.9_{\pm 2.5}$ | $9.6_{\pm 2.5}$ | $7.9_{\pm 2.0}$ |

Table 11: Certified accuracy (CA) [%] under joint $\ell_0$- and $\ell_1$-perturbations of categorical and numerical features, respectively, depending on whether model uses categorical and/or numerical features. The natural accuracy is the CA at radius $r = 0.0$. Larger is better.

| Dataset | Categorical Features | $\ell_0$ Radius $r_0$ | CA without Numerical Features | CA with Numerical Features at $\ell_1$ Radius $r_1$ | | | | | | |
|---|---|---|---|---|---|---|---|---|---|---|
| | | | | 0.00 | 0.25 | 0.50 | 0.75 | 1.00 | 1.25 | 1.50 |
| ADULT | no | - | - | $80.0_{\pm 0.3}$ | $77.3_{\pm 0.4}$ | $73.3_{\pm 0.4}$ | $69.8_{\pm 0.5}$ | $64.7_{\pm 0.6}$ | $61.5_{\pm 0.6}$ | $58.1_{\pm 0.7}$ |
| | yes | 0 | $69.7_{\pm 0.6}$ | $69.8_{\pm 0.6}$ | $69.2_{\pm 0.6}$ | $68.3_{\pm 0.6}$ | $67.4_{\pm 0.7}$ | $65.8_{\pm 0.6}$ | $64.7_{\pm 0.7}$ | $63.3_{\pm 0.6}$ |
| | | 1 | $52.0_{\pm 0.8}$ | $53.5_{\pm 0.7}$ | $52.9_{\pm 0.7}$ | $52.3_{\pm 0.7}$ | $51.5_{\pm 0.7}$ | $50.6_{\pm 0.8}$ | $49.7_{\pm 0.9}$ | $48.6_{\pm 0.9}$ |
| | | 2 | $27.5_{\pm 0.6}$ | $31.8_{\pm 0.5}$ | $30.7_{\pm 0.5}$ | $29.7_{\pm 0.4}$ | $28.9_{\pm 0.4}$ | $28.1_{\pm 0.5}$ | $27.6_{\pm 0.4}$ | $27.1_{\pm 0.5}$ |
| | | 3 | $6.6_{\pm 0.2}$ | $12.1_{\pm 0.3}$ | $10.8_{\pm 0.3}$ | $9.9_{\pm 0.2}$ | $9.1_{\pm 0.2}$ | $8.3_{\pm 0.2}$ | $8.0_{\pm 0.2}$ | $7.8_{\pm 0.1}$ |
| CREDIT | no | - | - | $69.8_{\pm 2.2}$ | $66.3_{\pm 2.0}$ | $62.7_{\pm 1.8}$ | $59.1_{\pm 0.9}$ | $53.3_{\pm 2.0}$ | $42.3_{\pm 1.6}$ | $35.1_{\pm 3.5}$ |
| | yes | 0 | $58.3_{\pm 9.5}$ | $58.0_{\pm 9.9}$ | $57.6_{\pm 9.8}$ | $57.0_{\pm 9.7}$ | $56.4_{\pm 9.5}$ | $55.8_{\pm 9.7}$ | $54.2_{\pm 10.5}$ | $53.6_{\pm 10.8}$ |
| | | 1 | $38.2_{\pm 7.4}$ | $39.0_{\pm 7.7}$ | $38.5_{\pm 7.7}$ | $38.3_{\pm 7.7}$ | $37.7_{\pm 7.7}$ | $37.2_{\pm 7.8}$ | $36.7_{\pm 7.8}$ | $35.5_{\pm 7.9}$ |
| | | 2 | $21.7_{\pm 5.0}$ | $22.4_{\pm 5.3}$ | $22.0_{\pm 5.0}$ | $21.4_{\pm 5.4}$ | $20.8_{\pm 5.7}$ | $20.6_{\pm 5.7}$ | $20.0_{\pm 6.0}$ | $19.1_{\pm 5.8}$ |
| | | 3 | $8.4_{\pm 2.6}$ | $9.5_{\pm 2.9}$ | $9.4_{\pm 2.7}$ | $8.9_{\pm 2.8}$ | $8.6_{\pm 2.9}$ | $8.3_{\pm 2.7}$ | $7.8_{\pm 2.9}$ | $7.5_{\pm 2.9}$ |
| MAMMO | no | - | - | $49.6_{\pm 3.7}$ | $48.0_{\pm 3.2}$ | $47.0_{\pm 3.3}$ | $47.0_{\pm 3.3}$ | $46.8_{\pm 3.1}$ | $44.6_{\pm 3.1}$ | $44.5_{\pm 3.1}$ |
| | yes | 0 | $79.0_{\pm 1.3}$ | $76.6_{\pm 1.7}$ | $76.5_{\pm 1.5}$ | $76.3_{\pm 1.6}$ | $76.3_{\pm 1.6}$ | $76.3_{\pm 1.6}$ | $74.2_{\pm 2.4}$ | $74.2_{\pm 2.4}$ |
| | | 1 | $31.8_{\pm 4.5}$ | $40.0_{\pm 5.4}$ | $38.5_{\pm 4.6}$ | $37.8_{\pm 4.7}$ | $37.8_{\pm 4.7}$ | $37.6_{\pm 4.7}$ | $36.1_{\pm 4.8}$ | $36.0_{\pm 5.0}$ |
| | | 2 | $0.0_{\pm 0.0}$ | $0.0_{\pm 0.0}$ | $0.0_{\pm 0.0}$ | $0.0_{\pm 0.0}$ | $0.0_{\pm 0.0}$ | $0.0_{\pm 0.0}$ | $0.0_{\pm 0.0}$ | $0.0_{\pm 0.0}$ |
| BANK | no | - | - | $80.4_{\pm 3.5}$ | $77.4_{\pm 3.0}$ | $75.1_{\pm 3.2}$ | $72.2_{\pm 3.1}$ | $68.4_{\pm 2.5}$ | $63.7_{\pm 2.2}$ | $57.0_{\pm 1.1}$ |
| | yes | 0 | $64.4_{\pm 7.5}$ | $74.0_{\pm 5.4}$ | $72.6_{\pm 5.1}$ | $71.6_{\pm 5.0}$ | $70.2_{\pm 5.0}$ | $68.3_{\pm 5.4}$ | $66.3_{\pm 5.4}$ | $63.2_{\pm 5.7}$ |
| | | 1 | $44.4_{\pm 7.9}$ | $56.5_{\pm 6.6}$ | $55.2_{\pm 6.3}$ | $53.9_{\pm 6.4}$ | $52.3_{\pm 6.3}$ | $50.2_{\pm 6.3}$ | $48.2_{\pm 6.2}$ | $44.9_{\pm 6.3}$ |
| | | 2 | $23.2_{\pm 7.1}$ | $38.5_{\pm 6.9}$ | $37.2_{\pm 6.5}$ | $36.0_{\pm 6.2}$ | $34.6_{\pm 5.9}$ | $32.8_{\pm 5.8}$ | $30.7_{\pm 5.7}$ | $27.7_{\pm 5.4}$ |
| | | 3 | $8.3_{\pm 4.7}$ | $20.2_{\pm 6.6}$ | $19.4_{\pm 6.4}$ | $18.4_{\pm 6.0}$ | $17.2_{\pm 5.6}$ | $16.0_{\pm 5.0}$ | $14.6_{\pm 4.6}$ | $12.3_{\pm 4.2}$ |

Table 12: Certified accuracy (CA) [%] and balanced certified accuracy (BCA) [%] under $\ell_0$-perturbations of categorical features. Larger is better.

| Dataset | $\ell_0$ Radius $r_0$ | CA | BCA |
|---|---|---|---|
| MUSHROOM | 0 | $90.6_{\pm 0.7}$ | $90.4_{\pm 0.9}$ |
| | 1 | $87.1_{\pm 1.7}$ | $86.9_{\pm 1.6}$ |
| | 2 | $81.2_{\pm 3.6}$ | $81.1_{\pm 3.4}$ |
| | 3 | $70.5_{\pm 5.7}$ | $70.7_{\pm 5.5}$ |

Table 13: We compare certifying the same stump ensembles via Deterministic Smoothing (DRS) and Randomized Smoothing (RS) with respect to the average certified radius (ACR) and the certified accuracy [%] at numerous radii $r$ on MNIST 1 vs. 5 for $\ell_1$ ($\lambda = 4.0$) and $\ell_2$ ($\sigma = 0.5$) norm perturbations. Larger is better.

| Norm | Method | ACR | Certified Accuracy at Radius r | | | | | | | |
|------|--------|-----|------|------|------|------|------|------|------|------|
| | | | 0.0 | 0.50 | 1.00 | 1.50 | 2.00 | 2.50 | 3.00 | 3.50 |
| $\ell_1$ | RS ($n = 100$) | 2.809 | 93.0 | 91.2 | 88.6 | 86.2 | 82.9 | 77.0 | 68.8 | 0.0 |
| | RS ($n = 1000$) | 3.337 | 95.6 | 94.4 | 92.8 | 90.6 | 87.8 | 84.7 | 79.5 | 70.4 |
| | RS ($n = 10000$) | 3.430 | 96.0 | 95.3 | 93.7 | 91.6 | 89.4 | 85.8 | 82.1 | 73.8 |
| | RS ($n = 100000$) | 3.456 | 96.1 | 95.5 | 94.0 | 91.9 | **89.9** | 86.3 | 82.9 | 74.6 |
| | DRS (ours) | **3.467** | **96.6** | **95.6** | **94.1** | **92.1** | **89.9** | **86.5** | **83.1** | **75.1** |
| $\ell_2$ | RS ($n = 100$) | 0.680 | 94.8 | 90.1 | 0.0 | 0.0 | 0.0 | 0.0 | 0.0 | 0.0 |
| | RS ($n = 1000$) | 1.102 | 95.6 | 92.5 | 85.0 | 0.0 | 0.0 | 0.0 | 0.0 | 0.0 |
| | RS ($n = 10000$) | 1.403 | 95.9 | 92.9 | 86.9 | 75.0 | 0.0 | 0.0 | 0.0 | 0.0 |
| | RS ($n = 100000$) | 1.627 | 95.9 | **93.0** | 87.3 | 78.1 | 0.0 | 0.0 | 0.0 | 0.0 |
| | DRS (ours) | **2.161** | **96.0** | **93.0** | **87.5** | **79.0** | **65.3** | **40.5** | **12.3** | **5.9** |

## C.2 Derandomized vs. Randomized Smoothing

Here, we compare evaluating stump ensembles deterministically via DRS (Section 3) to sampling-based RS [24]. In Table 13, we provide quantitative results corresponding to Fig. 6, expanded by an equivalent experiment for $\ell_1$-norm perturbations. We observe that as sampling-based RS uses increasingly more samples, it converges towards DRS. This convergence is much faster in the $\ell_1$-setting. However, especially in the $\ell_2$-setting, a notable gap remains even when using as many as $100\,000$ samples. This is expected as sampling-based RS computes a lower confidence bound to the true success probability, which can be computed exactly with DRS. Thus the higher the desired confidence, the larger this gap will be. Further, if RS were to yield a larger radius than DRS, this would actually be an error, occurring with probability $\alpha$, as DRS computes the true maximum certifiable radius. This highlights another key difference: RS provides probabilistic guarantees that hold with confidence $1 - \alpha$, while DRS provides deterministic guarantees. Moreover, for RS, many samples have to be evaluated (typically $n = 100\,000$), while DRS can efficiently compute the exact CDF. We note that the much larger improvement in certified radii observed for $\ell_2$-norm perturbations is due to the significantly higher sensitivity of the certifiably radius w.r.t. the success probability (see Table 1).

## C.3 MLE Optimality Criterion

In Table 14, we compare our robust MLE optimality criterion (MLE) to applying the standard entropy criterion to samples drawn from the input randomization scheme (Sampling) or the clean data (Default). We observe that training approaches accounting for randomness (i.e., Sampling and MLE) consistently outperform default training. In some cases, default training even suffers from a mode collapse, always predicting the same class. Amongst the two methods accounting for the input randomization, our MLE optimality criterion consistently outperforms samplings at all noise magnitudes and for both perturbation types. This effect is particularly pronounced at large noise magnitudes, where sampling becomes less effective at capturing the input distribution.

Table 14: We compare training stump ensembles optimally via MLE-optimal criterion, training them via noisy sampling (Sampling) and default training (Default) with respect to the average certified radius (ACR) and the certified accuracy [%] on MNIST 2 VS. 6 at numerous radii $r$ on various norms for multiple noise magnitudes ($\lambda$ for $\ell_1$ and $\sigma$ for $\ell_2$). Larger is better.

| Norm | $\lambda\,(\ell_1)$ or $\sigma\,(\ell_2)$ | Method | ACR | Certified Accuracy at Radius r | | | | | | | |
| | | | | 0.0 | 0.5 | 1.0 | 1.5 | 2.0 | 2.5 | 3.0 | 3.5 |
|---|---|---|---|---|---|---|---|---|---|---|---|
| $\ell_1$ | 1.0 | Default | 0.519 | 51.9 | 51.9 | 51.9 | 0.0 | 0.0 | 0.0 | 0.0 | 0.0 |
| | | Sampling | 0.928 | **96.2** | 93.9 | 64.8 | 0.0 | 0.0 | 0.0 | 0.0 | 0.0 |
| | | MLE (Ours) | **0.931** | 96.2 | **94.3** | **66.2** | 0.0 | 0.0 | 0.0 | 0.0 | 0.0 |
| | 4.0 | Default | 2.074 | 51.9 | 51.9 | 51.9 | 51.9 | 51.9 | 51.9 | 51.9 | 51.9 |
| | | Sampling | 3.166 | **96.3** | 95.0 | 93.3 | 90.5 | 87.3 | 81.4 | 72.5 | 56.0 |
| | | MLE (Ours) | **3.282** | **96.3** | **95.4** | **93.9** | **91.7** | **88.7** | **84.1** | **76.0** | **62.8** |
| | 16.0 | Default | 8.297 | 51.9 | 51.9 | 51.9 | 51.9 | 51.9 | 51.9 | 51.9 | 51.9 |
| | | Sampling | 6.646 | **96.4** | 95.3 | 94.4 | 93.4 | 91.8 | 90.0 | 87.8 | 84.9 |
| | | MLE (Ours) | **8.574** | 96.2 | **95.7** | **95.0** | **94.1** | **93.2** | **91.7** | **90.6** | **88.4** |
| $\ell_2$ | 0.25 | Default | 0.967 | 51.9 | 51.9 | 51.8 | 48.7 | 0.0 | 0.0 | 0.0 | 0.0 |
| | | Sampling | 1.628 | **96.3** | 92.8 | 85.9 | 71.7 | 0.0 | 0.0 | 0.0 | 0.0 |
| | | MLE (Ours) | **1.642** | **96.3** | **93.0** | **86.3** | **73.0** | 0.0 | 0.0 | 0.0 | 0.0 |
| | 1.0 | Default | **3.436** | 51.9 | 51.9 | 51.9 | 51.9 | **51.9** | **51.9** | **51.9** | **51.9** |
| | | Sampling | 1.594 | 95.5 | 89.1 | 76.5 | 57.9 | 33.5 | 11.7 | 2.1 | 0.2 |
| | | MLE (Ours) | 1.724 | 95.5 | **90.1** | **79.2** | **62.5** | 40.3 | 18.7 | 5.4 | 1.3 |
| | 4.0 | Default | **12.167** | 51.9 | 51.9 | 51.9 | 51.9 | **51.9** | **51.9** | **51.9** | **51.9** |
| | | Sampling | 1.095 | 89.2 | 72.9 | 50.9 | 32.6 | 15.8 | 4.0 | 0.5 | 0.0 |
| | | MLE (Ours) | 1.652 | **95.1** | **88.7** | **76.5** | **59.2** | 36.6 | 16.3 | 4.9 | 1.5 |

Table 15: Comparison of average certified radius (ACR) and certified accuracy at various radii $r$ with respect to the $\ell_1$ norm for numerous datasets and noise magnitudes $\lambda$. Larger is better.

| Dataset | $\lambda$ | ACR | Certified Accuracy at Radius r | | | | | | | | | | |
| | | | 0.0 | 1.0 | 2.0 | 3.0 | 4.0 | 5.0 | 6.0 | 7.0 | 8.0 | 9.0 | 10.0 |
|---|---|---|---|---|---|---|---|---|---|---|---|---|---|
| FMNIST-SHOES | 0.5 | 0.407 | 84.4 | 0.0 | 0.0 | 0.0 | 0.0 | 0.0 | 0.0 | 0.0 | 0.0 | 0.0 | 0.0 |
| | 1.0 | 0.766 | 83.5 | 55.1 | 0.0 | 0.0 | 0.0 | 0.0 | 0.0 | 0.0 | 0.0 | 0.0 | 0.0 |
| | 2.0 | 1.463 | 83.7 | 74.9 | 47.0 | 0.0 | 0.0 | 0.0 | 0.0 | 0.0 | 0.0 | 0.0 | 0.0 |
| | 4.0 | 2.780 | **85.8** | 80.2 | 73.3 | 60.9 | 21.3 | 0.0 | 0.0 | 0.0 | 0.0 | 0.0 | 0.0 |
| | 8.0 | 4.755 | 83.9 | 80.0 | 75.5 | 70.3 | 63.9 | 56.4 | 46.5 | 32.6 | 1.9 | 0.0 | 0.0 |
| | 16.0 | **7.975** | 84.3 | **81.7** | **77.8** | **75.0** | **71.7** | **67.3** | **63.2** | **57.9** | **52.9** | **47.0** | **41.1** |
| MNIST 1 VS. 5 | 0.5 | 0.476 | 96.3 | 0.0 | 0.0 | 0.0 | 0.0 | 0.0 | 0.0 | 0.0 | 0.0 | 0.0 | 0.0 |
| | 1.0 | 0.934 | 96.3 | 77.0 | 0.0 | 0.0 | 0.0 | 0.0 | 0.0 | 0.0 | 0.0 | 0.0 | 0.0 |
| | 2.0 | 1.808 | 96.2 | 92.1 | 62.8 | 0.0 | 0.0 | 0.0 | 0.0 | 0.0 | 0.0 | 0.0 | 0.0 |
| | 4.0 | 3.467 | 96.6 | 94.1 | 89.9 | 83.1 | 39.1 | 0.0 | 0.0 | 0.0 | 0.0 | 0.0 | 0.0 |
| | 8.0 | 6.472 | **97.0** | **95.4** | **93.3** | **91.0** | **87.4** | **82.2** | **75.1** | 60.7 | 4.4 | 0.0 | 0.0 |
| | 16.0 | **8.957** | 90.4 | 88.6 | 86.6 | 83.5 | 80.3 | 77.4 | 72.9 | **67.4** | **61.9** | **56.2** | **49.6** |
| MNIST 2 VS. 6 | 0.5 | 0.477 | 96.3 | 0.0 | 0.0 | 0.0 | 0.0 | 0.0 | 0.0 | 0.0 | 0.0 | 0.0 | 0.0 |
| | 1.0 | 0.931 | 96.2 | 66.2 | 0.0 | 0.0 | 0.0 | 0.0 | 0.0 | 0.0 | 0.0 | 0.0 | 0.0 |
| | 2.0 | 1.780 | 96.2 | 92.2 | 43.0 | 0.0 | 0.0 | 0.0 | 0.0 | 0.0 | 0.0 | 0.0 | 0.0 |
| | 4.0 | 3.282 | 96.3 | 93.9 | 88.7 | 76.0 | 3.8 | 0.0 | 0.0 | 0.0 | 0.0 | 0.0 | 0.0 |
| | 8.0 | 5.617 | **96.5** | 94.6 | 91.4 | 87.4 | 80.9 | 71.7 | 56.6 | 31.3 | 0.0 | 0.0 | 0.0 |
| | 16.0 | **8.574** | 96.2 | **95.0** | **93.2** | **90.6** | **86.5** | **82.7** | **77.5** | **70.5** | **62.7** | **53.3** | **41.3** |

## C.4 Effect of Noise Level

Here, we provide additional experiments for varying noise magnitudes, $\lambda$ for $\ell_1$-certification, and $\sigma$ for $\ell_2$-certification. In Tables 15 and 16, we provide extensive experiments for the $\ell_1$- and $\ell_2$-setting, respectively, which we visualize in Figs. 7 and 8.

We observe that, in the $\ell_1$-setting, the natural accuracy (certified accuracy at radius 0) is quite insensitive to an increase in noise magnitude. Consequently, large $\lambda$ lead to exceptionally large ACR and certified accuracies even at large radii, e.g., on MNIST 2 VS. 6, we obtain a certified accuracy of $82.7\%$ at $\ell_1$-radius $r = 5.0$.

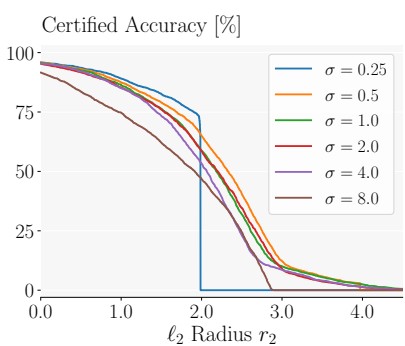

Figure 8: Comparing DRS for various noise levels $\sigma$ on MNIST 1 VS. 5.

Table 16: Comparison of average certified radius (ACR) and certified accuracy at various radii $r$ with respect to the $\ell_2$ norm for numerous datasets and noise magnitudes $\sigma$. Larger is better.

| Dataset | $\sigma$ | ACR | Certified Accuracy at Radius r | | | | | | | |
|---|---|---|---|---|---|---|---|---|---|---|
| | | | 0.0 | 0.5 | 1.0 | 1.5 | 2.0 | 2.5 | 3.0 | 3.5 |
| FMNIST-Shoes | 0.25 | 1.361 | **86.8** | **79.6** | 70.0 | **58.2** | 0.0 | 0.0 | 0.0 | 0.0 |
| | 0.5 | 1.723 | 86.5 | 78.9 | **70.1** | 56.6 | 42.2 | 27.8 | 17.4 | 8.4 |
| | 1.0 | 1.699 | 86.2 | 78.5 | 69.1 | 55.7 | 41.0 | 25.8 | 16.9 | 8.9 |
| | 2.0 | 1.681 | 86.2 | 78.5 | 68.8 | 55.1 | 40.2 | 25.6 | 16.8 | 8.7 |
| | 4.0 | **2.136** | 57.1 | 52.2 | 49.7 | 47.9 | **46.0** | **43.4** | **39.4** | **35.0** |
| | 8.0 | 1.518 | 83.7 | 74.2 | 64.4 | 51.0 | 35.4 | 21.0 | 10.4 | 4.8 |
| MNIST 1 vs. 5 | 0.25 | 1.737 | 95.8 | **93.6** | **89.0** | **82.8** | 0.0 | 0.0 | 0.0 | 0.0 |
| | 0.5 | **2.161** | 96.0 | 93.0 | 87.5 | 79.0 | **65.3** | **40.5** | 12.3 | 5.9 |
| | 1.0 | 2.044 | 96.0 | 92.7 | 86.1 | 75.6 | 57.3 | 28.4 | 11.4 | 6.7 |
| | 2.0 | 2.012 | 95.8 | 92.7 | 85.8 | 74.9 | 56.2 | 26.9 | 10.3 | 6.0 |
| | 4.0 | 1.875 | 94.8 | 87.2 | 71.8 | 48.1 | 34.7 | 29.9 | **23.8** | **15.7** |
| | 8.0 | 1.808 | **96.1** | 90.2 | 80.3 | 62.9 | 36.4 | 20.4 | 13.3 | 7.7 |
| MNIST 2 vs. 6 | 0.25 | 1.642 | **96.3** | **93.0** | **86.3** | **73.0** | 0.0 | 0.0 | 0.0 | 0.0 |
| | 0.5 | **1.824** | 95.8 | 91.2 | 81.9 | 66.7 | 46.4 | 23.4 | 7.4 | 1.3 |
| | 1.0 | 1.724 | 95.5 | 90.1 | 79.2 | 62.5 | 40.3 | 18.7 | 5.4 | 1.3 |
| | 2.0 | 1.688 | 95.4 | 89.5 | 78.0 | 60.9 | 38.8 | 17.5 | 4.9 | 1.0 |
| | 4.0 | 1.652 | 95.1 | 88.7 | 76.5 | 59.2 | 36.6 | 16.3 | 4.9 | 1.5 |
| | 8.0 | 1.718 | 74.3 | 61.0 | 53.2 | 49.4 | **46.6** | **40.2** | **30.3** | **17.4** |

In the $\ell_2$-setting, increasing the noise magnitude $\sigma$ generally leads to a more pronounced drop in natural and certified accuracy, and thus similar ACRs for various noise magnitudes.

**Thinking Outside the Box** Analysing this surprising behaviour in the $\ell_1$-setting, we empirically find that despite the data being normalized to $[0, 1]$, the MLE optimality criterion often yields split positions $v_m$ outside of $[0, 1]$. Recall that there, uniformly distributed random noise is added to the original sample ($\boldsymbol{x}' \sim Unif([\boldsymbol{x} - \lambda, \boldsymbol{x} + \lambda]^d)$). In Fig. 9, we show a histogram of the split positions ($v_m$), illustrating this behaviour. In the $\ell_1$-setting and using $\lambda = 2$, all split positions are either smaller than $-1$ or larger than 1.9, which are exactly the borders of uniform distributions with $\lambda = 2$ centered at the extremes of the image domain ($[0, 1]$). As all splits are outside the hyperbox constituting the original image domain, we refer to this behaviour as 'thinking outside the box'. Intuitively, each unperturbed data point is on the same side of $v_m$ in this case, but when

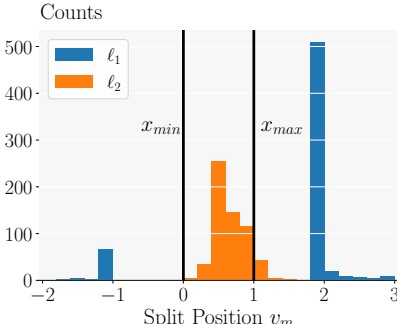

Figure 9: Comparing counts for values of $v_m$ on MNIST 2 vs. 6 for $\ell_1$ and $\ell_2$ norms with $\lambda = 2.0$ and $\sigma = 0.25$, respectively.

the randomization scheme is applied, a split outside of $[-1, 2]$ leads to a probability mass of 0 for an original feature value of 0 or 1, while for the other, the probability mass can be as high as $\frac{1}{2\lambda}$. Therefore, such splits allow the smoothed model to still separate these cases well for randomized inputs.

While we observe this effect on all datasets in the $\ell_1$-setting given a sufficiently large $\lambda$, it does not appear in the $\ell_2$-setting. There, $v_m$'s are typically clustered closely around or inside $[0, 1]$, as the Gaussian randomization applied here has unbounded support and does not permit for such a clean separation, regardless of the choice of $v_m$.

### C.5 Leaf Prediction Discretization

In the main paper, all experiments are conducted with leaf predictions discretized to $\Delta = 100$ values to enable our efficient CDF computation. In this section, we investigate the effect of this discretization. Concretely, we report results on MNIST 1 vs. 5 and BREASTCANCER using a range of discretization-granularities from 2 to 10 000 and 2 to 100 000 in Table 17 and Table 18, respectively. While using a very coarse discretization can lead to a mode collapse (explaining the very high ACRs observed for $\ell_2$ perturbations in Table 18) and generally degraded performance, we observe that for sufficiently fine discretizations (typically $\Delta \geq 50$) the results converge as the discretization is refined further. As the discretization effect on the ensemble's output is bounded by

Table 17: We compare the performance of models for different number of discretization sizes with respect to average certified radius (ACR) and given certified accuracies (CA) [%] on MNIST 1 vs. 5. We utilize $\lambda = 4.0$ for $\ell_1$ and $\sigma = 0.5$ for $\ell_2$. We report mean and standard deviation over 5-fold cross-validation. Larger mean is better.

| Norm | Discretizations | ACR | Certified Accuracy [%] at Radius r | | | | | | | |
|---|---|---|---|---|---|---|---|---|---|---|
| | | | 0.00 | 0.50 | 1.00 | 1.50 | 2.00 | 2.50 | 3.00 | 3.50 |
| $\ell_1$ | 2 | $1.860_{\pm 0.025}$ | $60.2_{\pm 1.3}$ | $52.9_{\pm 1.5}$ | $46.6_{\pm 1.3}$ | $44.5_{\pm 0.6}$ | $44.5_{\pm 0.6}$ | $44.5_{\pm 0.6}$ | $44.5_{\pm 0.6}$ | $44.3_{\pm 0.6}$ |
| | 3 | $2.220_{\pm 0.023}$ | $55.5_{\pm 0.6}$ | $55.5_{\pm 0.6}$ | $55.5_{\pm 0.6}$ | $55.5_{\pm 0.6}$ | $55.5_{\pm 0.6}$ | $55.5_{\pm 0.6}$ | $55.5_{\pm 0.6}$ | $55.5_{\pm 0.6}$ |
| | 5 | $2.220_{\pm 0.023}$ | $55.5_{\pm 0.6}$ | $55.5_{\pm 0.6}$ | $55.5_{\pm 0.6}$ | $55.5_{\pm 0.6}$ | $55.5_{\pm 0.6}$ | $55.5_{\pm 0.6}$ | $55.5_{\pm 0.6}$ | $55.5_{\pm 0.6}$ |
| | 10 | $2.191_{\pm 0.033}$ | $72.6_{\pm 1.2}$ | $68.1_{\pm 1.3}$ | $63.8_{\pm 1.0}$ | $58.6_{\pm 1.0}$ | $53.7_{\pm 0.8}$ | $48.6_{\pm 0.7}$ | $45.1_{\pm 0.6}$ | $44.3_{\pm 0.6}$ |
| | 50 | $3.502_{\pm 0.014}$ | $97.0_{\pm 0.4}$ | $96.2_{\pm 0.4}$ | $94.9_{\pm 0.5}$ | $93.2_{\pm 0.5}$ | $90.9_{\pm 0.5}$ | $88.0_{\pm 0.5}$ | $83.8_{\pm 0.4}$ | $76.9_{\pm 0.2}$ |
| | 100 | $3.425_{\pm 0.015}$ | $96.2_{\pm 0.4}$ | $94.7_{\pm 0.5}$ | $93.0_{\pm 0.5}$ | $91.0_{\pm 0.5}$ | $88.7_{\pm 0.5}$ | $85.7_{\pm 0.3}$ | $81.5_{\pm 0.4}$ | $74.2_{\pm 0.5}$ |
| | 500 | $3.375_{\pm 0.016}$ | $95.1_{\pm 0.5}$ | $93.6_{\pm 0.5}$ | $91.8_{\pm 0.5}$ | $89.8_{\pm 0.4}$ | $87.4_{\pm 0.3}$ | $84.4_{\pm 0.5}$ | $80.0_{\pm 0.5}$ | $72.5_{\pm 0.3}$ |
| | 1'000 | $3.367_{\pm 0.016}$ | $94.9_{\pm 0.6}$ | $93.5_{\pm 0.6}$ | $91.6_{\pm 0.5}$ | $89.7_{\pm 0.5}$ | $87.2_{\pm 0.3}$ | $84.2_{\pm 0.5}$ | $79.8_{\pm 0.6}$ | $72.3_{\pm 0.3}$ |
| | 5'000 | $3.364_{\pm 0.016}$ | $94.9_{\pm 0.6}$ | $93.4_{\pm 0.5}$ | $91.6_{\pm 0.5}$ | $89.6_{\pm 0.4}$ | $87.2_{\pm 0.3}$ | $84.2_{\pm 0.5}$ | $79.7_{\pm 0.5}$ | $72.4_{\pm 0.3}$ |
| | 10'000 | $3.364_{\pm 0.016}$ | $94.9_{\pm 0.6}$ | $93.4_{\pm 0.5}$ | $91.6_{\pm 0.5}$ | $89.6_{\pm 0.4}$ | $87.2_{\pm 0.3}$ | $84.2_{\pm 0.5}$ | $79.7_{\pm 0.6}$ | $72.3_{\pm 0.3}$ |
| $\ell_2$ | 2 | $1.766_{\pm 0.023}$ | $44.5_{\pm 0.6}$ | $44.5_{\pm 0.6}$ | $44.5_{\pm 0.6}$ | $44.5_{\pm 0.6}$ | $44.5_{\pm 0.6}$ | $44.5_{\pm 0.6}$ | $44.5_{\pm 0.6}$ | $44.5_{\pm 0.6}$ |
| | 3 | $2.204_{\pm 0.023}$ | $55.5_{\pm 0.6}$ | $55.5_{\pm 0.6}$ | $55.5_{\pm 0.6}$ | $55.5_{\pm 0.6}$ | $55.5_{\pm 0.6}$ | $55.5_{\pm 0.6}$ | $55.5_{\pm 0.6}$ | $55.5_{\pm 0.6}$ |
| | 5 | $2.204_{\pm 0.023}$ | $55.5_{\pm 0.6}$ | $55.5_{\pm 0.6}$ | $55.5_{\pm 0.6}$ | $55.5_{\pm 0.6}$ | $55.5_{\pm 0.6}$ | $55.5_{\pm 0.6}$ | $55.5_{\pm 0.6}$ | $55.5_{\pm 0.6}$ |
| | 10 | $2.044_{\pm 0.007}$ | $95.9_{\pm 0.5}$ | $92.2_{\pm 0.5}$ | $85.9_{\pm 0.2}$ | $75.8_{\pm 0.4}$ | $56.9_{\pm 1.1}$ | $27.8_{\pm 0.7}$ | $14.1_{\pm 0.4}$ | $7.6_{\pm 0.3}$ |
| | 50 | $2.110_{\pm 0.006}$ | $95.6_{\pm 0.6}$ | $92.0_{\pm 0.6}$ | $86.3_{\pm 0.3}$ | $77.7_{\pm 0.4}$ | $63.1_{\pm 0.6}$ | $38.0_{\pm 0.9}$ | $11.4_{\pm 0.3}$ | $5.4_{\pm 0.3}$ |
| | 100 | $2.120_{\pm 0.005}$ | $95.3_{\pm 0.5}$ | $91.8_{\pm 0.6}$ | $86.2_{\pm 0.4}$ | $77.7_{\pm 0.5}$ | $64.0_{\pm 0.7}$ | $39.8_{\pm 0.7}$ | $11.4_{\pm 0.3}$ | $5.0_{\pm 0.3}$ |
| | 500 | $2.125_{\pm 0.005}$ | $95.1_{\pm 0.5}$ | $91.8_{\pm 0.6}$ | $86.2_{\pm 0.3}$ | $77.8_{\pm 0.4}$ | $64.3_{\pm 0.6}$ | $40.7_{\pm 0.7}$ | $11.5_{\pm 0.4}$ | $4.8_{\pm 0.2}$ |
| | 1000 | $2.126_{\pm 0.005}$ | $95.1_{\pm 0.5}$ | $91.7_{\pm 0.6}$ | $86.2_{\pm 0.3}$ | $77.8_{\pm 0.4}$ | $64.4_{\pm 0.6}$ | $40.8_{\pm 0.7}$ | $11.6_{\pm 0.3}$ | $4.8_{\pm 0.2}$ |
| | 5'000 | $2.126_{\pm 0.005}$ | $95.1_{\pm 0.5}$ | $91.7_{\pm 0.6}$ | $86.2_{\pm 0.3}$ | $77.8_{\pm 0.4}$ | $64.3_{\pm 0.6}$ | $40.8_{\pm 0.8}$ | $11.6_{\pm 0.4}$ | $4.8_{\pm 0.2}$ |
| | 10'000 | $2.126_{\pm 0.005}$ | $95.1_{\pm 0.5}$ | $91.7_{\pm 0.6}$ | $86.2_{\pm 0.3}$ | $77.8_{\pm 0.4}$ | $64.3_{\pm 0.6}$ | $40.8_{\pm 0.8}$ | $11.6_{\pm 0.3}$ | $4.8_{\pm 0.2}$ |

Table 18: We compare the performance of models for different number of discretization sizes with respect to average certified radius (ACR) and given certified accuracies (CA) [%] on BREASTCANCER. We utilize $\lambda = 2.0$ for $\ell_1$ and $\sigma = 4.0$ for $\ell_2$. We report mean and standard deviation over 5-fold cross-validation. Larger mean is better.

| Norm | Discretizations | ACR | Certified Accuracy [%] at Radius r | | | | | |
|---|---|---|---|---|---|---|---|---|
| | | | 0.00 | 0.10 | 0.2 | 0.3 | 0.4 | 0.5 |
| $\ell_1$ | 2 | $1.396_{\pm 0.039}$ | $95.2_{\pm 1.4}$ | $94.1_{\pm 1.4}$ | $92.7_{\pm 1.4}$ | $90.6_{\pm 1.8}$ | $89.0_{\pm 1.9}$ | $87.1_{\pm 0.8}$ |
| | 3 | $1.298_{\pm 0.044}$ | $65.0_{\pm 2.2}$ | $65.0_{\pm 2.2}$ | $65.0_{\pm 2.2}$ | $65.0_{\pm 2.2}$ | $65.0_{\pm 2.2}$ | $65.0_{\pm 2.2}$ |
| | 5 | $1.292_{\pm 0.045}$ | $67.6_{\pm 2.8}$ | $66.9_{\pm 2.7}$ | $66.3_{\pm 2.3}$ | $65.9_{\pm 2.5}$ | $65.4_{\pm 2.0}$ | $65.0_{\pm 2.2}$ |
| | 10 | $1.395_{\pm 0.038}$ | $95.2_{\pm 1.4}$ | $94.1_{\pm 1.4}$ | $92.8_{\pm 1.2}$ | $90.8_{\pm 1.8}$ | $89.0_{\pm 1.9}$ | $87.1_{\pm 0.8}$ |
| | 50 | $1.396_{\pm 0.039}$ | $95.2_{\pm 1.4}$ | $94.1_{\pm 1.4}$ | $92.7_{\pm 1.4}$ | $90.6_{\pm 1.8}$ | $89.0_{\pm 1.9}$ | $87.1_{\pm 0.8}$ |
| | 100 | $1.396_{\pm 0.039}$ | $95.2_{\pm 1.4}$ | $94.1_{\pm 1.4}$ | $92.7_{\pm 1.4}$ | $90.6_{\pm 1.8}$ | $89.0_{\pm 1.9}$ | $87.1_{\pm 0.8}$ |
| | 500 | $1.396_{\pm 0.039}$ | $95.2_{\pm 1.4}$ | $94.1_{\pm 1.4}$ | $92.7_{\pm 1.4}$ | $90.6_{\pm 1.8}$ | $89.0_{\pm 1.9}$ | $87.1_{\pm 0.8}$ |
| | 1'000 | $1.396_{\pm 0.039}$ | $95.2_{\pm 1.4}$ | $94.1_{\pm 1.4}$ | $92.7_{\pm 1.4}$ | $90.6_{\pm 1.8}$ | $89.0_{\pm 1.9}$ | $87.1_{\pm 0.8}$ |
| | 5'000 | $1.396_{\pm 0.039}$ | $95.2_{\pm 1.4}$ | $94.1_{\pm 1.4}$ | $92.7_{\pm 1.4}$ | $90.6_{\pm 1.8}$ | $89.0_{\pm 1.9}$ | $87.1_{\pm 0.8}$ |
| | 10'000 | $1.396_{\pm 0.039}$ | $95.2_{\pm 1.4}$ | $94.1_{\pm 1.4}$ | $92.7_{\pm 1.4}$ | $90.6_{\pm 1.8}$ | $89.0_{\pm 1.9}$ | $87.1_{\pm 0.8}$ |
| | 50'000 | $1.396_{\pm 0.039}$ | $95.2_{\pm 1.4}$ | $94.1_{\pm 1.4}$ | $92.7_{\pm 1.4}$ | $90.6_{\pm 1.8}$ | $89.0_{\pm 1.9}$ | $87.1_{\pm 0.8}$ |
| | 100'000 | $1.396_{\pm 0.039}$ | $95.2_{\pm 1.4}$ | $94.1_{\pm 1.4}$ | $92.7_{\pm 1.4}$ | $90.6_{\pm 1.8}$ | $89.0_{\pm 1.9}$ | $87.1_{\pm 0.8}$ |
| $\ell_2$ | 2 | $20.672_{\pm 0.704}$ | $65.0_{\pm 2.2}$ | $65.0_{\pm 2.2}$ | $65.0_{\pm 2.2}$ | $65.0_{\pm 2.2}$ | $65.0_{\pm 2.2}$ | $65.0_{\pm 2.2}$ |
| | 3 | $1.533_{\pm 0.080}$ | $65.0_{\pm 2.2}$ | $65.0_{\pm 2.2}$ | $65.0_{\pm 2.2}$ | $65.0_{\pm 2.2}$ | $65.0_{\pm 2.2}$ | $65.0_{\pm 2.2}$ |
| | 5 | $1.533_{\pm 0.080}$ | $65.0_{\pm 2.2}$ | $65.0_{\pm 2.2}$ | $65.0_{\pm 2.2}$ | $65.0_{\pm 2.2}$ | $65.0_{\pm 2.2}$ | $65.0_{\pm 2.2}$ |
| | 10 | $20.672_{\pm 0.704}$ | $65.0_{\pm 2.2}$ | $65.0_{\pm 2.2}$ | $65.0_{\pm 2.2}$ | $65.0_{\pm 2.2}$ | $65.0_{\pm 2.2}$ | $65.0_{\pm 2.2}$ |
| | 50 | $0.644_{\pm 0.158}$ | $89.2_{\pm 1.2}$ | $80.5_{\pm 4.7}$ | $66.7_{\pm 16.8}$ | $59.8_{\pm 20.1}$ | $56.8_{\pm 18.2}$ | $54.5_{\pm 17.2}$ |
| | 100 | $0.653_{\pm 0.075}$ | $93.3_{\pm 5.7}$ | $90.6_{\pm 7.4}$ | $88.3_{\pm 8.2}$ | $85.2_{\pm 8.0}$ | $80.6_{\pm 7.9}$ | $73.8_{\pm 5.5}$ |
| | 500 | $0.624_{\pm 0.020}$ | $95.9_{\pm 1.7}$ | $93.9_{\pm 1.9}$ | $92.4_{\pm 2.5}$ | $89.3_{\pm 2.9}$ | $83.9_{\pm 1.8}$ | $77.3_{\pm 3.6}$ |
| | 1000 | $0.609_{\pm 0.015}$ | $96.1_{\pm 1.3}$ | $94.8_{\pm 1.8}$ | $92.7_{\pm 2.1}$ | $90.0_{\pm 1.3}$ | $85.9_{\pm 2.0}$ | $77.9_{\pm 2.2}$ |
| | 5'000 | $0.597_{\pm 0.015}$ | $96.7_{\pm 1.7}$ | $95.0_{\pm 1.7}$ | $92.8_{\pm 1.8}$ | $91.1_{\pm 1.4}$ | $86.3_{\pm 2.1}$ | $76.6_{\pm 3.5}$ |
| | 10'000 | $0.598_{\pm 0.017}$ | $96.5_{\pm 1.8}$ | $95.0_{\pm 1.4}$ | $92.8_{\pm 1.8}$ | $91.1_{\pm 1.4}$ | $86.3_{\pm 2.7}$ | $76.3_{\pm 3.1}$ |
| | 50'000 | $0.597_{\pm 0.016}$ | $96.7_{\pm 1.7}$ | $94.9_{\pm 1.6}$ | $92.8_{\pm 1.8}$ | $91.2_{\pm 1.6}$ | $86.2_{\pm 2.9}$ | $76.6_{\pm 3.5}$ |
| | 100'000 | $0.597_{\pm 0.016}$ | $96.7_{\pm 1.7}$ | $94.9_{\pm 1.6}$ | $92.8_{\pm 1.8}$ | $91.2_{\pm 1.6}$ | $86.2_{\pm 2.9}$ | $76.6_{\pm 3.5}$ |

Table 19: We compare the performance of models for different binning sizes with respect to average certified radius (ACR) and given certified accuracies (CA) [%] on MNIST 1 VS. 5. We utilize $\lambda = 4.0$ for $\ell_1$ and $\sigma = 0.5$ for $\ell_2$. Larger is better.

| Norm | Binning Size | ACR | Certified Accuracy [%] at Radius r | | | | | | | |
|------|------|------|------|------|------|------|------|------|------|------|
| | | | 0.00 | 0.50 | 1.00 | 1.50 | 2.00 | 2.50 | 3.00 | 3.50 |
| $\ell_1$ | 4.0 | 3.452 | 96.2 | 95.2 | 93.6 | 91.7 | 89.4 | 86.3 | 82.7 | 75.3 |
| | 2.0 | 3.452 | 96.2 | 95.2 | 93.6 | 91.7 | 89.4 | 86.3 | 82.7 | 75.3 |
| | 1.0 | 3.468 | 96.5 | 95.6 | 94.1 | 92.0 | 89.9 | 86.5 | 83.1 | 75.2 |
| | 0.5 | 3.465 | 96.6 | 95.5 | 94.2 | 91.9 | 89.9 | 86.6 | 83.1 | 75.0 |
| | 0.1 | 3.462 | 96.6 | 95.5 | 94.2 | 92.0 | 89.9 | 86.4 | 83.0 | 74.8 |
| | 0.05 | 3.466 | 96.5 | 95.6 | 94.1 | 91.9 | 89.9 | 86.5 | 83.1 | 75.1 |
| | 0.01 | 3.467 | 96.6 | 95.6 | 94.1 | 92.1 | 89.9 | 86.5 | 83.1 | 75.1 |
| | 0.005 | 3.467 | 96.6 | 95.6 | 94.1 | 92.1 | 89.9 | 86.5 | 83.1 | 75.1 |
| | 0.001 | 3.467 | 96.5 | 95.6 | 94.1 | 92.1 | 90.0 | 86.5 | 83.1 | 75.1 |
| | 0.0005 | 3.466 | 96.5 | 95.5 | 94.1 | 92.1 | 90.0 | 86.5 | 83.0 | 75.1 |
| | 0.0001 | 3.467 | 96.5 | 95.5 | 94.2 | 92.1 | 89.9 | 86.5 | 83.1 | 75.1 |
| $\ell_2$ | 4.0 | 0.584 | 56.0 | 55.9 | 31.5 | 0.0 | 0.0 | 0.0 | 0.0 | 0.0 |
| | 2.0 | 1.888 | 95.4 | 91.3 | 83.6 | 73.9 | 55.7 | 22.4 | 4.0 | 0.8 |
| | 1.0 | 1.980 | 95.7 | 92.0 | 84.1 | 74.0 | 58.8 | 35.2 | 4.8 | 0.6 |
| | 0.5 | 2.119 | 95.8 | 93.0 | 87.2 | 78.5 | 62.6 | 35.7 | 11.5 | 6.5 |
| | 0.1 | 2.161 | 96.0 | 93.1 | 87.5 | 79.1 | 65.2 | 40.8 | 12.3 | 5.9 |
| | 0.05 | 2.160 | 96.0 | 93.1 | 87.5 | 79.1 | 65.3 | 41.4 | 12.4 | 5.8 |
| | 0.01 | 2.161 | 96.0 | 93.0 | 87.5 | 79.0 | 65.3 | 40.5 | 12.3 | 5.9 |
| | 0.005 | 2.163 | 96.0 | 93.1 | 87.5 | 79.0 | 65.3 | 40.9 | 12.4 | 5.9 |
| | 0.001 | 2.164 | 96.0 | 93.0 | 87.5 | 79.0 | 65.3 | 41.1 | 12.5 | 5.8 |
| | 0.0005 | 2.164 | 96.0 | 93.0 | 87.5 | 79.0 | 65.3 | 41.2 | 12.5 | 5.8 |
| | 0.0001 | 2.163 | 96.0 | 93.0 | 87.5 | 79.0 | 65.3 | 41.1 | 12.5 | 5.8 |

$\frac{M}{2\Delta}$, we conclude that these fine discretizations closely approximate the non-discretized case. We choose $\Delta = 100$ such that our discretized smoothed models generally approximately recover the behavior of the non-discretized models while allowing for fast computations of the ensemble PDF.

While performance improves monotonically with finer discretizations for MNIST 1 VS. 5 in the $\ell_2$ setting and for BREASTCANCER in the $\ell_1$ setting, it seems to peak and then declines again for MNIST 1 VS. 5 in the $\ell_1$ setting and for BREASTCANCER in the $\ell_2$ setting. For BREASTCANCER, we observe significantly larger standard deviations at coarse discretizations leading to overlapping $\pm 1$ standard deviation intervals for all discretization levels not suffering from a mode collapse and thus statistically insignificant results. For MNIST 1 VS. 5 in the $\ell_1$, the performance peak at $\Delta = 50$ is statistically significant. We hypothesize that the coarser regularizations have a beneficial regularizing effect in this setting.

## C.6 Split Position Search Granularity

In our main paper, all experiments are conducted using a step size of $0.01$ to conduct the line search for the optimal split position $v_m$. In Table 19, we report results for search granularities from $4.0$ to $10^{-4}$ and observe that a step size of $0.1$ is sufficiently fine and reducing it further does not improve the performance of the obtained models. This suggest that our approximate optimization based on line search comes very close to the finding the true optimal split position and thus jointly MLE optimal $v_m$ and $\gamma$.

## C.7 Error Bounds

In Table 20 we report the mean and standard deviation of the certified accuracies at various radii across a 5-fold cross validation for datasets including only numerical features. We observe, that our results are very stable with standard deviations of less than $1.0\%$ on the computer vision datasets, which have large sample sizes. On the tabular datasets, which consist of much fewer samples, the dependence on the train/test-split is slightly larger with standard deviations reaching around $4.0\%$ in some settings. Only where a large noise magnitude ($\sigma = 4$), very small sample sizes, and large radii come together (BREASTCANCER for $\ell_2$ perturbations of $r = 1.0$) do we observe a large sensitivity to the train/test-split and thus a high standard deviations of up to $21\%$.

Table 20: Average certified accuracy (ACR) Certified accuracy (CA) [%] at various radii with respect to $\ell_1$- and $\ell_2$-norm bounded perturbations on various datasets. Larger is better.

| Perturbation | Dataset | ACR | Radius $r$ | | | | |
|---|---|---|---|---|---|---|---|
| | | | 0.00 | 0.10 | 0.25 | 0.50 | 1.00 |
| $\ell_1$ | BREASTCANCER | $1.396_{\pm 0.039}$ | $95.2_{\pm 1.4}$ | $94.1_{\pm 1.4}$ | $92.1_{\pm 1.3}$ | $87.1_{\pm 0.8}$ | $72.8_{\pm 2.8}$ |
| | DIABETES | $0.153_{\pm 0.010}$ | $73.7_{\pm 3.1}$ | $58.3_{\pm 2.9}$ | $30.1_{\pm 4.0}$ | $0.0_{\pm 0.0}$ | $0.0_{\pm 0.0}$ |
| | SPAMBASE | $2.541_{\pm 0.042}$ | $89.1_{\pm 0.5}$ | $88.2_{\pm 0.4}$ | $87.4_{\pm 0.6}$ | $85.2_{\pm 0.8}$ | $80.6_{\pm 1.2}$ |
| | FMNIST-SHOES | $2.731_{\pm 0.027}$ | $84.4_{\pm 0.8}$ | $83.8_{\pm 0.8}$ | $83.1_{\pm 0.8}$ | $81.7_{\pm 0.8}$ | $79.1_{\pm 0.7}$ |
| | MNIST 1 VS. 5 | $3.425_{\pm 0.015}$ | $96.2_{\pm 0.4}$ | $95.9_{\pm 0.5}$ | $95.4_{\pm 0.5}$ | $94.7_{\pm 0.5}$ | $93.0_{\pm 0.5}$ |
| | MNIST 2 VS. 6 | $3.243_{\pm 0.008}$ | $95.7_{\pm 0.3}$ | $95.5_{\pm 0.3}$ | $95.1_{\pm 0.3}$ | $94.5_{\pm 0.3}$ | $92.8_{\pm 0.3}$ |
| $\ell_2$ | BREASTCANCER | $0.653_{\pm 0.075}$ | $93.3_{\pm 5.7}$ | $90.6_{\pm 7.4}$ | $87.3_{\pm 8.6}$ | $73.8_{\pm 5.5}$ | $15.5_{\pm 21.5}$ |
| | DIABETES | $0.124_{\pm 0.005}$ | $72.7_{\pm 3.5}$ | $53.0_{\pm 3.4}$ | $15.6_{\pm 2.6}$ | $0.0_{\pm 0.0}$ | $0.0_{\pm 0.0}$ |
| | SPAMBASE | $0.884_{\pm 0.006}$ | $89.7_{\pm 1.0}$ | $87.4_{\pm 1.2}$ | $83.7_{\pm 1.1}$ | $73.6_{\pm 1.0}$ | $40.9_{\pm 0.8}$ |
| | FMNIST-SHOES | $1.334_{\pm 0.012}$ | $85.0_{\pm 0.8}$ | $83.7_{\pm 0.9}$ | $81.5_{\pm 0.7}$ | $78.1_{\pm 0.5}$ | $68.6_{\pm 0.7}$ |
| | MNIST 1 VS. 5 | $1.720_{\pm 0.006}$ | $95.3_{\pm 0.4}$ | $94.8_{\pm 0.4}$ | $94.0_{\pm 0.5}$ | $92.3_{\pm 0.6}$ | $87.9_{\pm 0.3}$ |
| | MNIST 2 VS. 6 | $1.613_{\pm 0.007}$ | $95.5_{\pm 0.2}$ | $94.9_{\pm 0.3}$ | $93.9_{\pm 0.2}$ | $91.7_{\pm 0.2}$ | $84.9_{\pm 0.7}$ |

# D   (De-)Randomized Smoothing for Decision Tree Ensembles

While we focus on decision stump ensembles in the main paper and in particular in Section 3, our approach can easily be extended to ensembles of decision trees with arbitrary depths which do not use the same features in distinct decision trees. In particular, our approach can be easily extended to arbitrary individual decision trees.

Recall that the key idea of our approach is to group individual decision stumps into independent meta-stumps, allowing us to represent the output of the overall smoothed ensemble as the sum of independent terms. We can apply the same idea here by constructing meta-stumps over all features used in an individual tree. As we do not permit features to be reused in multiple trees, every tree is independent of all others.

For every leaf $j$ of a decision tree $m$ with prediction $\gamma_{m,j}$, we can accumulate the constraints along the path from the root to the leaf of that tree as

$$\psi_j(\boldsymbol{x}) = \bigwedge_i x_i > v_{j,i}^- \wedge x_i \le v_{j,i}^+. \tag{11}$$

Note that if $x_i$ is not constrained (in one direction) on the path to leaf $j$, we can simply set the corresponding threshold $v^{\{+,-\}}$ to $\pm\infty$. This allows us to formally define a smoothed decision tree as

$$g_m(\boldsymbol{x}) = \sum_j \gamma_{m,j} \prod_i \mathbb{P}_{\boldsymbol{x}_i' \sim \phi(\boldsymbol{x}_i)}[v_{j,i}^- < x_i' \le v_{j,i}^+]. \tag{12}$$

As all features are perturbed independently under the randomization scheme $\phi$, we can compute the probability of a perturbed sample satisfying $\psi_j$ and thus landing in leaf $j$ by factorization as

$$\mathbb{P}_{\boldsymbol{x}' \sim \phi(\boldsymbol{x})}[\psi_j(\boldsymbol{x}')] = \prod_i \underbrace{\mathbb{P}_{\boldsymbol{x}' \sim \phi(\boldsymbol{x})}[x_i' > v_{j,i}^- \wedge x_i' \le v_{j,i}^+]}_{p_{j,i}} \tag{13}$$

$$= \prod_i \mathbb{P}_{\boldsymbol{x}' \sim \phi(\boldsymbol{x})}[x_i' \le v_{j,i}^+] - \mathbb{P}_{\boldsymbol{x}' \sim \phi(\boldsymbol{x})}[x_i' \le v_{j,i}^-]. \tag{14}$$

For many common randomization schemes where the (dimension-wise) CDF is available, this expression can be evaluated efficiently in closed form, e.g., when using a Gaussian distribution as the randomization scheme $\phi(\boldsymbol{x}) = \mathcal{N}(\boldsymbol{x}, \sigma\boldsymbol{I})$, typically used for $\ell_2$-norm certificates (see Table 1), and given the inverse Gaussian CDF $\Phi^{-1}$, we obtain

$$\mathbb{P}_{\boldsymbol{x}' \sim \phi(\boldsymbol{x})}[\psi_j(\boldsymbol{x}')] = \prod_i \Phi^{-1}\left(\frac{v_{j,i}^+ - x_i}{\sigma}\right) - \Phi^{-1}\left(\frac{v_{j,i}^- - x_i}{\sigma}\right). \tag{15}$$

We can now construct a meta-stump equivalent per decision tree, where the piece-wise constant regions with output $\gamma_{m,j}$ are now simply defined over multiple variables instead of over a single

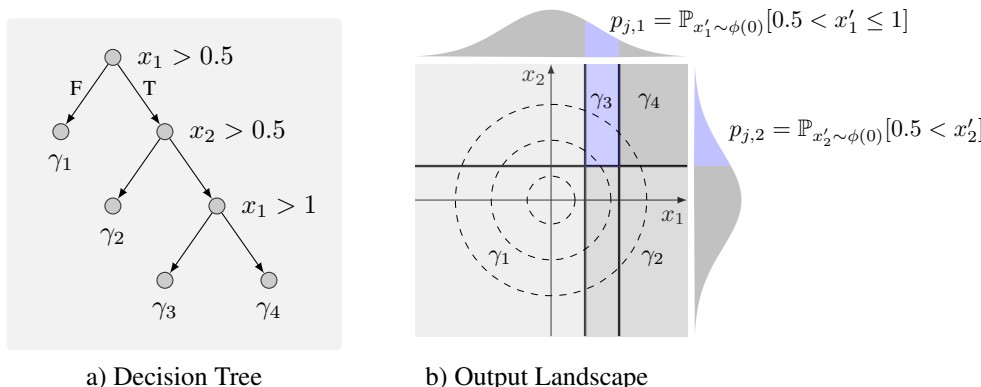

a) Decision Tree    b) Output Landscape

Figure 10: Illustration of a meta-stump on a decision tree with feature re-use.

variable. We illustrate this in Fig. 10, where we show a decision tree (Fig. 10a)) on features $x_1$ and $x_2$ (partially truncating depth to avoid clutter) and the corresponding output landscape (Fig. 10b)). We can now compute the probability of $\boldsymbol{x}' \sim \phi(\boldsymbol{0})$ falling into the blue region as the product of the probabilities of $x_1'$ lying in $[0.5, 1)$, $p_{j,1}$, and $x_2'$ lying in $[0.5, \infty)$, $p_{j,2}$. Proceeding similarly for the other regions, we can instantiate Algorithm 1, as for regular meta-stumps, only replacing the probability computation as discussed above.

This allows us to adapt Algorithm 1 to iterate over the ensembled decision trees instead of over features.