# OpenReview forum: "(De-)Randomized Smoothing for Decision Stump Ensembles"
_NeurIPS.cc/2022/Conference — NeurIPS 2022 Accept_

### Official Review · Reviewer_TjF1 · 2022-06-24

**Rating:** 5
**Confidence:** 5
**Soundness:** 3 good
**Presentation:** 3 good
**Contribution:** 2 fair

**Summary:**

This paper deterministic smoothing for decision stump ensembles, and derive an MLE-optimal training method for smoothed decision stumps under randomization.

**Questions:**

1. The necessity of using randomized smoothing to provide the robustness guarantee for such a simple tree-based classifier, Decision Stump Ensembles. I think this paper might solve a simple problem (endowing Decision Stump Ensemble with certified robustness) in an overly complex way (randomized smoothing).

2. Is the comparison to [Wang et al. 2020] in Table 2 is fair?  I think comparing smoothed decision stump ensemble (linear classifier) to naive decision stump ensemble (non-linear classifier) is unfair. It is unclear whether the improvement is for the superior performance of non-linear classifier or the certification method.


[Wang et al. 2020] On lp-norm robustness of ensemble decision stumps and trees，ICML 2020.

**Ethics Review Area:**

["I don’t know"]

**Limitations:**

The contribution of this paper is incremental. The paper just applies randomized smoothing to decision stump ensemble, and derives the robustness certificate for such a special classifier. The key contribution, deterministic certificate, is because of the overly simple structure of the base classifier (decision stump ensemble), instead of some new smoothing distributions (e.g., splitting noise [Levine et al., 2021]). We can also easily compute deterministic certificates for some other simple classifiers. For instance, we can derive the deterministic certificates for linear classifiers. The reason why we compute the probabilistic certificates for randomized smoothing is, the complex structure (almost black-box) of the neural network classifier. Moreover, I think the randomized smoothing for decision stump ensemble is only a simplified verison of the general problems studied in [Cohen et al., 2019] [Yang et al., 2020].

[Levine et al., 2021] Improved, Deterministic Smoothing for L_1 Certified Robustness, ICML 2021.

[Cohen et al., 2019] Certified adversarial robustness via randomized smoothing, ICML 2019.

[Yang et al., 2020] Randomized smoothing of all shapes and sizes, ICML 2020.


**Strengths And Weaknesses:**

Strengths:
1. Good presentation

Weaknesses:

1.  The necessity of using randomized smoothing to provide the robustness guarantee for Decision Stump Ensemble.

2. The contribution is incremental.

---

> ### Author Response · Authors · 2022-08-02
> **Response to Reviewer TjF1**
>
> $\newcommand{Ro}{\textcolor{blue}{K1dy}}$
> $\newcommand{Rt}{\textcolor{green}{7Kxj}}$
> $\newcommand{Rth}{\textcolor{red}{TjF1}}$
>
> We thank the reviewer $\Rth$ for appreciating the presentation of our work and address their comments below:
>
> **Q 3.1: Is (De-)Randomized Smoothing necessary to provide robustness guarantees for such simple architectures like ensembles of decision stumps?**
>
> A: As shown by Wang et al. 2020, the complete certification of decision stump ensembles is NP-complete for the specifications we consider ($\ell_1$- and $\ell_2$-norm balls). Therefore, incomplete methods such as ours or the one proposed by Wang et al. 2020 are needed for verification. As our (De-)Randomized Smoothing yields up to 4-times larger certified accuracies than the current state-of-the-art, we believe that it is perfectly suitable for the problem we tackle. Further, while ensembles of decision stumps might be a relatively simple architecture, they are still highly non-linear and practically relevant for many real-world applications, as pointed out by $\Rt$.
>
> **Q 3.2: Is the comparison of “linear” smoothed decision stump ensembles to the “non-linear” decision stump ensembles of Wang et al. 2020 in Table 2 fair?**
>
> A: First, we want to highlight that both our smoothed and the “naive” decision stump ensembles of Wang et al. 2020 are non-linear classifiers. We assume that the reviewer refers to the difference in behavior for inputs that are close to the threshold of some decision stumps. While “naive” stumps behave as step functions, smoothed stumps interpolate between the output values on the two sides of the threshold. This difference in behavior does not necessarily increase the performance of the resulting classifier, as can be seen in the updated Table 2 where we show that standard training for “naive” stump ensembles leads to a similar or higher natural accuracy than we obtain for smoothed stump ensembles. Further, given that we compare models with an equal number of parameters and very similar architectures, we believe that if smoothed stump ensembles perform significantly better than naive stump ensembles while retaining all their advantages, this would make our work even more valuable to real-world practitioners.
>
> **Q 3.3: Is the contribution of the paper merely an application of Randomized Smoothing (RS) and thus incremental?**
>
> A: No. As was well summarized by $\Ro$, our key contributions lie in leveraging the special structure of decision stump ensembles to construct an efficient (De-)Randomized Smoothing approach, overcoming the main disadvantages of RS and enabling our MLE-optimal training method.  These contributions allow us to obtain an up to 4-fold increase in certified accuracy compared to prior work, which we strongly believe to be substantial progress and not merely incremental. While it is indeed trivial to construct a randomized smoothing based deterministic verifier for linear classifiers, we want to highlight that (smoothed) decision stump ensembles are highly non-linear and that constructing a suitable (De-)Randomized Smoothing approach requires two key insights: i) grouping (dependent) decision stumps using the same feature into independent meta-stumps allows to efficiently compute their output distributions, and ii) a dynamic programming approach enables the efficient aggregation of the output distributions of individual (meta-)stumps into the overall ensemble PDF. Given these observations, we believe neither the structure of stump ensembles to be “overly simple” nor our approach a “simplified version” of general RS.
>
>
> **Q 3.4: The soundness of the paper seems “poor”.**
>
> A: We would like to ask the reviewer to explain his concerns regarding the soundness of our approach and/or evaluation, as we could not find any corresponding comment in the text of the review. In particular, we would like to point the reviewer to Theorem 3.1 and the corresponding proof in A.1, which shows the correctness of our method.
>
>
> We hope to have addressed the reviewer’s concerns, would like to strongly encourage them to update their review in accordance with the rating descriptions or more clearly state which “major technical flaws” they see in our work and are happy to answer any further questions.

---

> > ### Comment · Reviewer_TjF1 · 2022-08-06
> > **The essence of this paper is not deterministic randomized smoothing. Actually, randomized smoothing strongly limits the generality of this paper!**
> >
> > In this paper, the meaning of randomized smoothing is to convert the decision stump (**a piecewise constant function**) $f(x)=\gamma_l+ (\gamma_r -\gamma_r) 1_{x>v}$ to **a composition of the random disribution's PDF** $ f(x)=\gamma_l \mathcal{P}_{x' \sim \phi(x)}[x'<v] + \gamma_r \mathcal{P}_{x' \sim \phi(x)} [x' > v]$. However, we can totally neglect randomized smoothing and simply convert  the decision stump to other non-linear functions $f(x)= \gamma_l f_l(x) + \gamma_r f_r(x)$ (e.g., $f_l(x), f_r(x)$ can be customized by the user) to obtain stronger robustness. Meanwhile, the certified robustness can be easily computed when $f_l(x), f_r(x)$ are not so complex.  In summary, I feel that leveraging randomized smoothing is a bit redundant.  Randomized smoothing actually restricts the expression of $f_l(x), f_r(x)$ instead.
> >
> > Randomized smoothing is a useful certified defense for complex black-box models, since the robustness certificate of randomized smoothing do not rely on any property of the base classifier. However, for those fully explainable classifiers, I believe there exist many other effective defenses that leverage the property of the base classifier to obtain stronger robustness. In this paper, the author leverages the simple architecture of the decision stump to save the computation cost of Monte Carlo sampling, and derive deterministic certificates.

---

> > > ### Author Response · Authors · 2022-08-06
> > > **Response to Reviewer TjF1**
> > >
> > > $\newcommand{Rth}{\textcolor{red}{TjF1}}$
> > > $\newcommand{Rt}{\textcolor{green}{7Kxj}}$
> > >
> > > We thank the reviewer $\Rth$ for engaging in the discussion and increasing their soundness score from 1 to 3, contribution score from 1 to 2, and their overall evaluation from 2 to 4.
> > >
> > > When considering a single decision stump, we indeed replace the piecewise constant function $f(x) = \gamma_l + (\gamma_r - \gamma_l) 1_{x>v}$ with a smooth interpolation between the two values $f(x) = \gamma_l P_{x’\sim\phi(x)}[x’ \leq v]+ \gamma_r P_{x’\sim\phi(x)}[x’ > v]$.
> > > However, we respectfully disagree with the conclusions $\Rth$ draws from this. In particular, we identify and address the following 3 conclusions (C1-C3): $\Rth$ suggests that, we can (C1) simply replace $P_{x’\sim\phi(x)}[x’ \leq v]$ and $P_{x’\sim\phi(x)}[x’ > v]$ with other non-linear functions $f_l(x)$ and $f_r(x)$, disconnecting our approach from RS; (C2) easily compute certified robustness for less complex $f_l(x)$ and $f_r(x)$; and (C3) thus obtain stronger robustness certificates.
> > >
> > > **C1: We can “totally neglect randomized smoothing” (RS) and replace $P_{x’\sim\phi(x)}[x’ \leq v]$ and $P_{x’\sim\phi(x)}[x’ > v]$ with other non-linear functions $f_l(x)$ and $f_r(x)$.**
> > >
> > > A: First, our robustness guarantees are directly based on RS and thus this definition of $f_l(x)$ and $f_r(x)$. Replacing them with arbitrary non-linear functions and thus disconnecting our approach from (De-)Randomized Smoothing would require a completely new derivation of robustness certificates (see C2). Second, such a replacement would fundamentally change the architecture we analyze. However, as decision stump ensembles have been shown to be particularly effective in many important domains (as recognized by $\Rt$), we believe their analysis to be highly relevant.
> > >
> > > **C2: For “not so complex” $f_l(x)$ and $f_r(x)$, “certified robustness can be easily computed”.**
> > >
> > > A: We respectfully disagree. In fact, Wang et al. have shown that even for the simple class of piecewise constant functions $f_l$ and $f_r$ (i.e., standard decision stump ensembles), this problem is NP-complete. Therefore, we believe that choosing (other) non-linear $f_l$ and $f_r$ such that this problem becomes easy to solve is actually highly non-trivial. Note that $\ell_\infty$-certification is indeed easily feasible for piecewise linear functions, however, we consider the more challenging $\ell_1$ and $\ell_2$-settings.
> > >
> > > **C3: Customizing $f_l$ and $f_r$ would allow us “to obtain stronger robustness”.**
> > >
> > > A: First, we point out that our DRS approach already obtains much stronger robustness guarantees (up to 4-times larger certified accuracies) than the current state-of-the-art for tree-based models. Second, we want to highlight that decision stump ensembles can represent compositions of arbitrary univariate piecewise constant functions and thus approximate a very wide class of functions. Third, in many domains, the core challenge (especially for certified robustness) is not the expressivity of the model, but learning its parametrization from limited data, which we tackle with our MLE optimality criterion (Section 4 and Appendix A.2-A.4). Finally, we highlight that our (De-)Randomized Smoothing approach allows for the derivation of certificates against different attacker models simply by changing the randomization scheme, an advantage that would be lost using “customized” $f_l$ and $f_r$.
> > >
> > > **Leveraging the problem structure**
> > >
> > > We agree with $\Rth$ that while standard RS is a powerful technique for obtaining robustness certificates for black-box models, it can be improved by leveraging the structure of the underlying model. In fact, this is exactly what we do. We leverage the structure of stump ensembles to first construct meta-stumps and then deterministically compute their output distribution, to not only save the computation cost of Monte Carlo sampling (yielding multiple orders of magnitude speed-up and enabling our MLE-optimal training) but also to obtain deterministic certificates. Both are significant improvements and are acknowledged by $\Rth$. In addition, we obtain much larger (average) certified radii than standard RS in some settings (see Figure 6 and Section 5.3), which $\Rth$ suggests to be desirable.
> > > Extending this idea of leveraging model structure to improve upon the black-box case in more complex settings (beyond the tree ensembles discussed in the new Appendix D) is an interesting and promising direction for future work.
> > >
> > > To summarize, we strongly believe that Randomized Smoothing is not limiting the generality of our paper but instead forms the very foundation of our work, enabling the efficient training via our MLE optimality criterion and the derivation of deterministic certificates.
> > >
> > > We hope to have addressed all of the reviewer’s concerns and are happy to discuss further.

---

> > > > ### Comment · Reviewer_TjF1 · 2022-08-07
> > > > **Randomized smoothing is unnecessary**
> > > >
> > > > A) "First, we point out that our DRS approach already obtains much stronger robustness guarantees (up to 4-times larger certified accuracies) than the current state-of-the-art for tree-based models. "
> > > >
> > > > I admit that DRS greatly improves the robustness. However, the improvement is at the cost of its preferable architecture (step function). I suggest the author compare DRS with other non-step-function-architecture defenses.
> > > >
> > > > B) I summarize the contribution of DRS into:
> > > >
> > > > 1) Replace the piecewise constant function with $f(x)=\gamma_l \mathcal{P}_{x' \sim \phi(x)}[x'<v] + \gamma_r \mathcal{P}_{x' \sim \phi(x)} [x' > v]$.
> > > >
> > > > 2) RS is a tool to help us derive the robustness certificate.
> > > >
> > > > Why we should understand such replacement 1) in the way of randomized smoothing? It just replaces the decision stump with a non-linear function with some preferable properties (e.g. we can have the closed-form robustness certificates). Since we can write the closed-form expression of the smoothed function, we can disconnect randomized smoothing. Randomized smoothing would only limit your choices of $f_l(x), f_r(x)$.
> > > >
> > > > C)  "Second, such a replacement would fundamentally change the architecture we analyze. However, as decision stump ensembles have been shown to be particularly effective in many important domains"
> > > >
> > > > I believe that randomized smoothing also fundamentally changes the architecture.
> > > >
> > > > D) "we highlight that our (De-)Randomized Smoothing approach allows for the derivation of certificates against different attacker models simply by changing the randomization scheme, an advantage that would be lost using “customized” "
> > > >
> > > > I admit that different randomization schemes can generate different $f_l(x), f_r(x)$. However, the number randomization schemes that have the closed-form certificates is relatively limited, only including cubical distributions, spherical distributions, cross polytope distributions, log concave distributions. See [Yang2020Randomized]. The expression forms of smoothed stump decision are limited.  Therefore, I strongly suggest the author jump out randomized smoothing and consider the solution from a high level.
> > > >
> > > > [Yang2020Randomized] Randomized Smoothing of All Shapes and Sizes, Yang, ICML 2020.

---

> > > > > ### Author Response · Authors · 2022-08-07
> > > > > **Response to Reviewer TjF1**
> > > > >
> > > > > $\newcommand{Rt}{\textcolor{green}{7Kxj}}$
> > > > > $\newcommand{Rth}{\textcolor{red}{TjF1}}$
> > > > > We thank reviewer $\Rth$ for the quick reply, lively discussion, and again increasing their overall score.
> > > > >
> > > > > **Architecture (A,C)**
> > > > > $\Rth$ suggests, (C) that Randomized Smoothing fundamentally changes the underlying architecture and (A) that this change is undesirable in the case of decision stump ensembles.
> > > > > While we agree that RS does change the analyzed model (see section 7), we believe this change to be much more nuanced than completely replacing the underlying architectures, as it preserves many model properties such as interpretability. For example, the effect of changing an input feature of a smoothed decision stump ensemble can still be easily understood and isolated. For numeric features, the continuous output change of the smoothed ensemble is perhaps even more intuitive than the abrupt change observed for standard stump ensembles upon reaching some threshold value.
> > > > > To us, this interpretability in addition to the small number of samples required for training are the main reasons for choosing tree-based models. Therefore, we believe a comparison within this model class, in contrast to, e.g., a comparison with black-box neural networks, to be appropriate.
> > > > >
> > > > > **Contribution (B)**
> > > > > $\Rth$ claims that the main contribution of DRS is to replace individual (piecewise constant) decision stumps with the smooth $f(x)=\gamma_l \mathcal{P}{x' \sim \phi(x)}[x'<v] + \gamma_r \mathcal{P}{x' \sim \phi(x)} [x' > v]$. We want to highlight that i) such a direct replacement is only possible if every feature is used by at most one decision stump, otherwise, our proposed aggregation to meta-stumps is required, ii) given such an independent representation per input feature, computing the ensemble output distribution is a non-trivial problem solved by our efficient dynamic programming approach, and iii) training a stump ensemble that actually performs well under input noise is a challenge, we tackle successfully using our MLE optimality criterion.
> > > > >
> > > > > **Randomization Schemes (D)**
> > > > > While we agree with $\Rth$ that not all randomization schemes are amenable to our DRS approach, those, to the best of our knowledge, commonly used in Randomized Smoothing are.
> > > > >
> > > > > **Future Work**
> > > > > We agree with $\Rth$ that investigating the properties of our smoothed models independently from Randomized Smoothing with the goal of developing a generalized and more flexible formulation is a very exciting item for future work. However, especially given the empirical effectiveness of our DRS approach, we believe the present work to be an important stepping stone for such an extensive follow-up investigation, which we would like to share with the community.
> > > > >
> > > > > We again thank $\Rth$ for the interesting discussion and for suggesting such an interesting future work item. We hope to have addressed all remaining concerns and are happy to discuss further.

---

> ### Author Response · Authors · 2022-08-04
> **Re: Score Increase**
>
> $\newcommand{Rth}{\textcolor{red}{TjF1}}$
> We were delighted to see that reviewer $\Rth$ updated their soundness score from 1 to 2 and their overall evaluation from 2 to 3.
> If any soundness concerns regarding our approach remain, we would like to ask the reviewer to clarify the points in question, such that we can address them within the discussion period.

---

### Official Review · Reviewer_7Kxj · 2022-07-11

**Rating:** 7
**Confidence:** 3
**Soundness:** 3 good
**Presentation:** 2 fair
**Contribution:** 3 good

**Summary:**

The paper focuses on the problem of certifying ensembles of decision stumps efficiently and deterministically. By leveraging discretization and piecewise-constant nature of decision stumps, the paper presents a way to compute the output distribution (PDF and CDF) efficiently in a closed form for standard input randomizations without requiring any sampling. This paper also discusses the joint certification over numerical and categorical features and demonstrates how the proposed scheme can be extended to handle categorical features. Consequently, using the deterministic certification scheme, the paper proposes a novel way to construct robust decision tree stumps that can then be used to build ensembles through bagging or gradient boosting and adaptive boosting. The empirical evaluations demonstrate the improved robust predictive performance of the proposed schemes over existing baselines and the ability of the proposed schemes to provide joint certificates.


**Questions:**

- One question is if the authors can provide some intuition behind the dynamic programming based update in Algorithm 1. For example, why would the update of $\text{\tt pdf}[i][t + \Gamma_{ij}]$ need $\text{\tt pdf}[i-1][t]$? This could use a better explanation.

- In the learning via minimizing $H_{\text{entropy}}$ for $v_m$, it seems that the values $p^y$ would need to be constantly updated when performing the line search. Is that correct? This is not clear from the notation.

- The overall algorithm relies on the discretization level $\Delta$ (selected as 100). What is effect of this size of discretization $\Delta$ on the natural accuracy or certified accuracy? There must be a non-trivial tradeoff since a granular discretization would probably lead to a very noisy pdf while a coarser discretization would lead to a smoother one. Moreover, does the best level of discretization depend on the data (and if so how) or is a choice of discretization quite stable across datasets?


- How is the proposed PDF computation extensible to arbitrary decision trees? It is mentioned in the discussion of limitations.


**Limitations:**

The authors explicitly address limitations and potential negative societal impact. One of the limitations (being unable to "handle arbitrary ensembles of decision trees") is related to the weakness I listed above. The authors claim that it is possible to extend to decision trees, but the extension seems non-trivial. However, the authors are upfront with this limitation.


**Strengths And Weaknesses:**

Strengths:

- This paper handles the robustness certification of a very widely used class of models -- ensembles of decision tree (stumps) -- and hence has the potential of significant impact given the wide use of such models in practical applications.

- The paper provides a concise but sufficient background on randomized smoothing that allows the reader to understand the challenges that the proposed de-randomized smoothing is able circumvent.

- It is very intuitive how the authors leverage the piecewise-constant structure in decision stump ensembles and the input randomization (independent across dimensions) to efficiently compute the output distribution. This is a great example of leveraging structure in the problem for efficient and accurate solutions. This efficient solution also allows the authors to develop a novel constructiion of robust decision tree stumps which would not have been possible without such an efficient computation of the output distribution.


Weaknesses:

- One weakness of the proposed scheme is the focus on decision stumps, which facilitates the creation of the per-feature meta-stumps, that are critical in the efficient computation. It is not clear how this technique would transfer to deeper decision trees since it is not clear to me how the meta-stumps could be created for such trees.

---

> ### Author Response · Authors · 2022-08-02
> **Response to Reviewer 7Kxj**
>
> $\newcommand{Ro}{\textcolor{blue}{K1dy}}$
> $\newcommand{Rt}{\textcolor{green}{7Kxj}}$
> $\newcommand{Rth}{\textcolor{red}{TjF1}}$
>
> We thank the reviewer $\Rt$ for their insightful questions and helpful suggestions and are happy that they appreciate the significant impact, novelty, ingenuity, and presentation of our method. Below we answer their remaining questions:
>
> **Q 2.1: Can you provide some intuition behind the dynamic programming based update in Algorithm 1?**
>
> A: Yes. The field $\texttt{pdf}[i][t]$ in the dynamic programming table corresponds to the probability of the smoothed classifier returning the prediction $t$ (before normalization and in discretization steps) when considering only the first $i$ features or rather their corresponding meta-stumps. Given the pdf after considering $i-1$ features $\texttt{pdf}[i-1][*]$, we compute $\texttt{pdf}[i][t]$ as follows: Intuitively, after considering the $i^{th}$ feature the models output is $t$ exactly if after $i-1$ features it would have been $t-\Gamma_i$ and the $i^{th}$ meta stump predicts $\Gamma_i$. As the $i^{th}$ meta stump might return a range of $\Gamma_{i,j}$, we sum over the probabilities of obtaining $t-\Gamma_{i,j}$ after $i-1$ features weighted by the probability of the $i^{th}$ meta stump predicting $\Gamma_{i,j}$. We have extended the corresponding section to provide this intuition also in the paper, visualize it for 3 meta-stumps in Figure 1 and provide a correctness proof in Appendix A.1.
>
> **Q 2.2: Do the $p_{i,j}$, used during training, need to be recomputed for every threshold $v_m$?**
>
> A: Yes, this is correct. However, for the commonly used randomization schemes (e.g., Laplacian or Gaussian noise) this boils down to one evaluation of the univariate distribution’s CDF, which can be easily parallelized across all samples in the training set. We did not explicitly highlight this dependence in the notation to avoid clutter and have updated the corresponding section to make explicit note of this fact.
>
> **Q 2.3: What is the effect of the number of discretization steps on the (certified) accuracy? Would finer discretizations lead to very noisy PDFs?**
>
> A: We investigated this on the example of MNIST 1 vs. 5 (Table 15) in Appendix C.5, where we evaluate our model using $\Delta \in [2,1000]$. While we observe that very coarse discretizations can have a strong effect, results are consistent for sufficiently fine discretizations ($\Delta=50$). We picked $\Delta=100$ as empirically fine enough to recover the continuous behavior and did not observe a trade-off between natural and certified accuracy.
> While a very fine discretization would indeed lead to a very noisy PDF, we generally only use the CDF (both in training and certification), which is much less impacted and becomes smoother with a finer discretization. We have now conducted an additional experiment on the tabular data set Breast Cancer (see Table 16 in the updated Appendix C.5). We observe very similar trends, with the exact $\Delta$ required to recover the continuous behavior varying slightly with setting ($\ell_2$ vs. $\ell_1) and dataset.
>
> **Q 2.4: How can the PDF computation be extended to arbitrary decision trees and ensembles of specific decision trees?**
>
> A: We have added a detailed description of this extension as Appendix D and provide an intuition here. The key idea of our work is to represent the output of the smoothed (stump) ensemble as the sum of independent terms by grouping conditionally dependent (under the input randomization) decision stumps into independent meta stumps. We can apply the same idea to ensembles of trees as long as any one feature is used by at most one tree. To this end, we construct meta-stump equivalents per tree over all features used by this tree. Instead of having piecewise constant regions over a single variable, we now define them over multiple variables. By aggregating all constraints along the path from root to leaf over the same feature, we can factor the probability of reaching this specific leaf into independent terms allowing for an efficient computation, which has linear complexity in the number of leaves and the number of features used by the tree. For a more detailed discussion, please see the new Appendix D.
> We focused on decision stump ensembles because they empirically perform significantly better than (ensembles of) decision trees in the $\ell_p$ robustness setting we consider ( Wang et al. 2020). We hypothesize that this is because for stump ensembles, the manipulation of a single feature can only affect the prediction of one (meta) stump, while for tree ensembles, it can affect every tree using that feature.
>
> We hope to have addressed all of the reviewer’s questions and concerns, thank the reviewer again for suggesting to include the extension to decision trees in Appendix D, and are happy to answer any follow-up questions.

---

> > ### Comment · Reviewer_7Kxj · 2022-08-07
> > **Thank you for the response. Few follow-ups.**
> >
> > I thank the authors for the detailed response. Thank you for the intuition in Q2.1.
> >
> > - Regarding Q2.3, I am unable to find the results mentioned in the latest revision of the supplementary material in Tables 15 & 16. I wonder if the authors mean Tables 17 and 18. The results do seem to indicate (to me at least) that:
> >   - (a) Too coarse nor too fine a discretization can lead to degradation in performance since both ACR and certified accuracy at radius $r$ seems to increase and then decrease (although at a slower rate but still significantly) in many cases, implying that discretization does play some role.
> >   - (b) The best empirical level of discretization seems to somewhat vary between datasets and also within datasets for different norms ($\ell_1$ vs $\ell_2$).
> >   - (c) The ACR for BreastCancer (Table 18) look a bit odd, and could use some explanation.
> >   - (d) The appendix C.5 mentions a "non-discretized" baseline. Can you please elaborate and explain how we are able to solve the non-discretized version.
> >
> > - Regarding Q2.4, thank you for the details in the supplement and the high level explanation. We are able to leverage the condition that  *" any one feature is used by at most one tree"* and get the independence we need to create the per-feature meta-stumps. The novelty here is now being able to handle multiple features in the same tree by partitioning the space into (a small number of)  piece-wise constant regions. I am assuming that this will be somewhat manageable as long as the tree is not too deep otherwise the number of partitions will end up being very large and the meta-stump computation would require us to keep around a large table, right?

---

> > > ### Author Response · Authors · 2022-08-08
> > > **Reply to Follow-ups**
> > >
> > > $\newcommand{Rt}{\textcolor{green}{7Kxj}}$
> > >
> > > We thank reviewer $\Rt$ for engaging in the discussion, are happy that we could clarify most points already, and answer follow-up questions below:
> > >
> > > **Regarding Q2.3:**
> > >
> > > * Indeed, we meant Tables 17 and 18, apologies for the inconvenience.
> > > * (a and b) We agree that the discretization level does play some role and could have a regularizing effect. However, rather than a trade-off between accuracy and robustness, we typically observe uniform performance changes.
> > > While too coarse discretizations lead to severe degradation of performance (up to a full mode collapse), finer discretizations (above some threshold) typically have only a minor effect. We agree that this threshold depends on the exact setting considered. In more detail: While performance improves slowly but monotonically for MNIST $\ell_2$ and Breast Cancer $\ell_1$ with finer discretizations, there seemed to be a performance peak at some discretization level for MNIST $\ell_1$ and Breast Cancer $\ell_2$. We have now conducted a 5-fold cross-validation on these experiments (see updated Tables 17 and 18) and observe that for Breast Cancer $\ell_2$ the standard deviation at coarser discretization levels is significantly larger, such that the $\pm 1$ standard deviation ranges overlap for all settings not experiencing a mode collapse.  For $\ell_1$ perturbations on MNIST, the best performance is obtained with $\Delta = 50$ and worsens (statistically significantly) for finer discretizations. We think that this is an interesting observation and believe this might be due to a regularizing effect of the coarser discretization. While tuning the discretization level for every setting could yield slight performance improvements, we refrain from doing so and chose $\Delta=100$ with the intent to approximate the non-discretized model.
> > > * (c) The very large ACRs for some coarse discretizations are due to a partial or full mode collapse. Depending on whether the underlying (non-smoothed) model always (with probability at least $1-\epsilon$ with $\epsilon ~10^{-16}$) or almost always (with probability at least $0.5$) outputs the same class, we obtain very large ACRs of up to $\sim 20.67$. In this case, the certified radii are limited by our (double precision) floating point computations and could theoretically be infinite. We remark that for such settings, ACR is not a meaningful measure and we only report it for completeness.
> > > * (d) We did indeed not compute the non-discretized version, as this would require exponential time and memory in the number of considered stumps. When remarking that a sufficiently fine discretization could (approximately) recover the “non-discretized” behavior, we relied on the observation that the model behavior converges for increasingly finer discretizations, approximating the “non-discretized” behavior increasingly well as the discretization error is bounded by $\frac{M}{2\Delta}$, where $M$ is the number of considered stumps. We added experiments considering a larger $\Delta$ range of up to $100\,000$ (see updated Tables 17 and 18) and now observe this convergence in all settings. We have updated the text in Appendix C5 to reflect this better.
> > >
> > > **Regarding Q.2.4:**
> > >
> > > Indeed, the number of piecewise constant regions is equivalent to the number $n$ of leaves of a tree. While dense/unpruned trees have an exponential number of leaves in the tree depth $d$, thus limiting our method to moderately deep trees, typical trees often have much fewer leaves, allowing deeper trees to be considered. Computing the probability of selecting any specific such leaf is only order $\mathcal{O}(d)$ per input sample. Further, our DP table has the same size as for a stump ensemble with the same number of stumps as we have trees, as the number of rows corresponds to the number of (meta) ensemble members $M$ and the number of columns to the discretized output range, which is $\Delta M$ regardless of whether we use stumps or trees. Note that we actually only ever keep the two most recent rows in memory and that the smaller number of ensemble members $M$ actually reduces the size of these rows when considering tree ensembles. Updating the DP table for tree ensembles works very similarly to Algorithm 1, with only the number of piecewise constant regions increased from $M_i = 2$ for a standard stump to $M_i = n$ for our meta-stump equivalents. We further note that some of this increase in computational cost is compensated by the much smaller ensemble size as every feature can only be part of one tree.
> > >
> > > We hope to have addressed the remaining questions and are happy to discuss further.

---

### Official Review · Reviewer_K1dy · 2022-07-11

**Rating:** 6
**Confidence:** 2
**Soundness:** 3 good
**Presentation:** 3 good
**Contribution:** 3 good

**Summary:**

This work tries to produce robust decision stump ensembles against feature perturbation. To achieve this, the authors propose DeRandomized Smoothing (DRS), which can help provide a robust certificate. Moreover, the authors propose a robust MLE optimality criterion for individual stumps and two boosting schemes for whole ensembles. The experimental results look promising and demonstrate the effectiveness of the proposed method.

**Questions:**

1. Since you already have the code and the pipeline, do you mind comparing the baseline [23] and your proposed method on more tabular data during the rebuttal? Can you produce a table similar to Table 2 for datasets "mammo", "mushroom", "bank marketing", and "spambase"?

2. How did you do the train-test set split? Have you performed 5-fold cross validation? Could you conduct 5-CV on Table 2 and report the mean and standard deviation of both the baseline [23] and your proposed method?

**Limitations:**

The authors have addressed the limitations in Section 7.

**Strengths And Weaknesses:**

Strength:

1. The writing and figure presentation are very good.
2. The PDF computation via dynamic programming looks novel.
3. There are substantial improvements on experimental results, and the code is attached as part of the submission.

Weakness:

1. One part I would like to see better is I wish the authors could conduct experiments comparing the baseline [23] and the proposed work on more tabular data. Breastcancer and Diabetes seem not sufficient to me in Table 2. Moreover, results for the baseline [23] are directly copied from [23]. I would like to see more experimental results on more datasets to assess the effectiveness of the proposed method.

2. The other thing I'm concerned about is that all results reported in Table 2 have no standard deviations. It is not clear for me how the training and test set split was done or whether there is any overfitting.

---

> ### Author Response · Authors · 2022-08-02
> **Response to Reviewer K1dy**
>
> $\newcommand{Ro}{\textcolor{blue}{K1dy}}$
> $\newcommand{Rt}{\textcolor{green}{7Kxj}}$
> $\newcommand{Rth}{\textcolor{red}{TjF1}}$
>
> We thank the reviewer $\Ro$ for their helpful suggestions and are happy that they appreciate the effectiveness, novelty, and presentation of our method. Below we answer their remaining questions:
>
> **Q 1.1: Can you extend the evaluation of your method to further datasets, and compare to Wang et al. 2020 on them?**
>
> A: Yes. We conducted additional experiments on all the suggested tabular datasets and report these results (including statistics over cross validation; see Q1.2 below) in Appendix C.1 and in the new Appendix C.7.
> Unfortunately, we could not reproduce the results reported by Wang et al. as they neither provide the relevant models nor the exact hyperparameter choices. Further, we were unable to recover their results by training new models using the hyperparameters suggested in their README and following the guidelines in their paper combined with some manual hyperparameter tuning. To still enable a fair comparison, we considered exactly their setting (train/test split and certified radii) and report their results directly.  For this reason, we have also not added the new results to Table 2, instead reporting them in Appendix C.1 and C.7. Below, we show a subset of the hyperparameter settings, we evaluated and the obtained results.
> Note also, that given the training set, the training of our stump ensembles is fully deterministic, thus we do not report a standard deviation, when comparing to prior work on a fixed train/test-split.
>
> **Breast cancer dataset, $\ell_1$-norm (reproduction from Wang et al.):**
>
> | | Natural Accuracy (NAC) [\%] | Certified Accuracy (CA) at radius 1.0 [\%] |
> |---|---|---|
> |reported|98. 5|64.2|
> |---|---|---|
> |sl=2,lr=0.2| 74.5 | 74.5 |
> |sl=4,lr=0.2| 74.5 | 74.5 |
> |sl=6,lr=0.2| 83.9 | 73.0 |
> |sl=8,lr=0.2| 74.5 | 74.5 |
> |sl=10,lr=0.2| 90.5 | 45.3 |
> |---|---|---|
> |sl=2,lr=0.4| 74.5 | 74.5|
> |sl=4,lr=0.4 | 76.6 | 74.5 |
> |sl=6,lr=0.4| 95.6 | 35.0 |
> |sl=8,lr=0.4| 74.5 | 74.5 |
> |sl=10,lr=0.4| 100 | 51.8 |
> |---|---|---|
> |sl=2,lr=0.6| 74.5 | 74.5 |
> |sl=4,lr=0.6| 91.2 | 74.5 |
> |sl=6,lr=0.6| 100 | 52.5 |
> |sl=8,lr=0.6| 74.5 | 74.5 |
> |sl=10,lr=0.6| 100 | 32.8 |
>
> **Q 1.2: Can you clarify how the test/train-split was chosen and report cross validation results?**
>
> A: We discuss the choice of test/train-split in detail in Appendix B.1 and generally followed previous work in our choice to enable a direct comparison. For the datasets used in Table 2, we followed the splits from Wang et al. 2020, as we were unable to reproduce their results (see discussion in Q1.1). We have now conducted additional experiments using 5-CV on the newly suggested as well as all previously included datasets, and report results in Appendix C.1 and C.7 (Tables 8-11 and Table 20). We generally observe results consistent with Table 2, and especially the large computer vision data sets show only minimal variance. Only where a large noise magnitude $\sigma=4$, very small sample sizes, and large radii came together (Breast Cancer for $\ell_2$ perturbations of $r=1.0$) did we observe a large sensitivity to the train/test-split and thus high standard deviations.
>
> We hope to have addressed all of the reviewer’s questions and concerns and are happy to answer any follow-up questions.

---

### Author Response · Authors · 2022-08-02
**General Response**

$\newcommand{Ro}{\textcolor{blue}{K1dy}}$
$\newcommand{Rt}{\textcolor{green}{7Kxj}}$
$\newcommand{Rth}{\textcolor{red}{TjF1}}$

We thank the reviewers for their insightful questions, interesting comments, and helpful suggestions. We are happy to hear that they appreciate the significant impact ($\Rt$), ingenuity ($\Rt$), effectiveness ($\Ro$), novelty ($\Ro$,$\Rt$), and presentation($\Ro$,$\Rt$,$\Rth$) of our method.
We address the reviewers’ questions in individual replies and have updated the submitted and updated our paper and appendix (in the supplementary file) to incorporate suggestions and reflect clarifications.

We hope to have addressed the reviewers' questions and are happy to answer any follow-up questions or comments and look forward to the reviewers’ replies.

---

### Meta-Review · Area_Chair_rjHt · 2022-08-31

**Recommendation:** Accept
**Confidence:** Less certain

**Metareview:**

There were substantial discussions around this paper and its contributions. The authors did a good job of explaining and interacting with reviewers (with, as a consequence, a substantial raise of scores). To prepare the camera-ready version, it is strongly suggested to include the material introduced at discussion time, including the experimental results (K1dy) and use the intuitive explanation provided for the dynamic programming approach (7Kxj) to revamp a section / paragraph on a high-level explanation of the approach.

**Award:**

No

---

### Decision · Program_Chairs · 2022-09-14

Accept